# Patch n' Pack: NaViT, a Vision Transformer for any Aspect Ratio and Resolution

**Mostafa Dehghani**[*], **Basil Mustafa**[*], **Josip Djolonga**[†], **Jonathan Heek**[†],
**Matthias Minderer, Mathilde Caron, Andreas Steiner, Joan Puigcerver,**
**Robert Geirhos, Ibrahim Alabdulmohsin, Avital Oliver, Piotr Padlewski,**
**Alexey A. Gritsenko, Mario Lucic, Neil Houlsby**
Google DeepMind
{dehghani,basilm}@google.com

## Abstract

The ubiquitous and demonstrably suboptimal choice of resizing images to a fixed resolution before processing them with computer vision models has not yet been successfully challenged. However, models such as the Vision Transformer (ViT) offer flexible sequence-based modeling, and hence varying input sequence lengths. We take advantage of this with NaViT (Native Resolution ViT) which uses sequence packing during training to process inputs of arbitrary resolutions and aspect ratios. Alongside flexible model usage, we demonstrate improved training efficiency for large-scale supervised and contrastive image-text pretraining. NaViT can be efficiently transferred to standard tasks such as image and video classification, object detection, and semantic segmentation and leads to improved results on robustness and fairness benchmarks. At inference time, the input resolution flexibility can be used to smoothly navigate the test-time cost-performance trade-off. We believe that NaViT marks a departure from the standard, CNN-designed, input and modelling pipeline used by most computer vision models, and represents a promising direction for ViTs.

## 1   Introduction

The simple, flexible and scalable nature of the Vision Transformer (ViT) [1] has rendered it an almost ubiquitous replacement to convolution based neural networks. Underpinning this model is a simple operation: splitting an image into patches, each of which is linearly projected to a token. Typically, input images are resized to a fixed square aspect ratio and then split into a fixed number of patches.

Recent works have explored alternatives to this paradigm: FlexiViT [2] supports multiple patch sizes within one architecture, enabling smooth variation of sequence length and thus compute cost. This is achieved via random sampling of a patch size at each training step and a resizing algorithm to allow the initial convolutional embedding to support multiple patch sizes. Pix2Struct [3] introduced an alternative patching approach which preserves the aspect ratio, which is particularly useful for tasks such as chart and document understanding.

We present an alternative, NaViT. Multiple patches from different images are packed in a single sequence—termed Patch n' Pack—which enables variable resolution while preserving the aspect

---

[*] Project lead. [†]Core contributor.

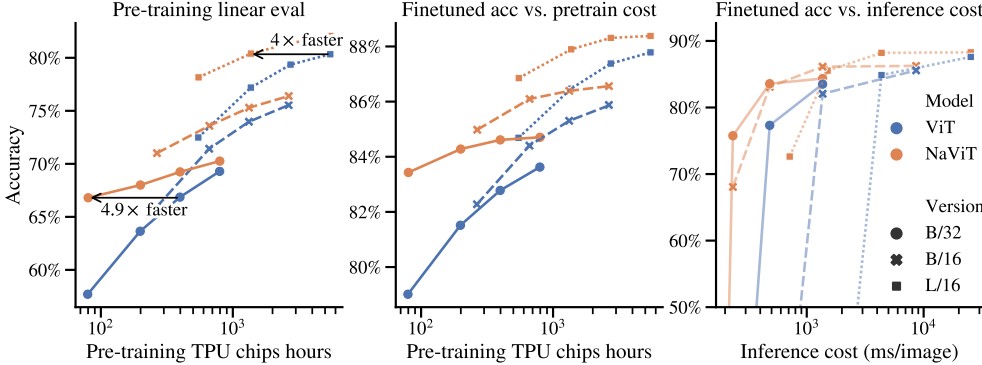

**Figure 1:** NaViT offers notable computational efficiency during pre-training (left) which carries over to downstream fine-tuning (middle). A single NaViT can be applied successfully to multiple resolutions (right), smoothly trading off performance and inference cost.

ratio (Figure 2). This is inspired by example packing in natural language processing, where multiple examples are packed into a single sequence to accommodate efficient training on variable length inputs.

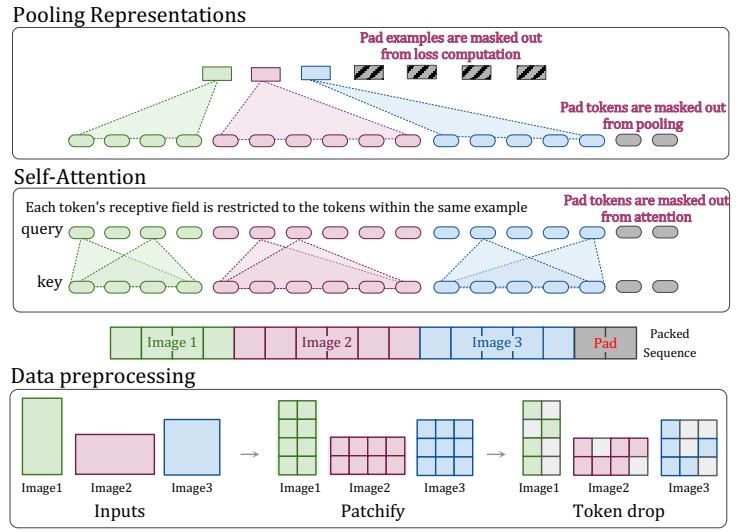

**Figure 2:** Example packing enables variable resolution images with preserved aspect ratio, reducing training time, improving performance and increasing flexibility. We show here the aspects of the data preprocessing and modelling that need to be modified to support Patch n' Pack. The position-wise operations in the network, such as MLPs, residual connections, and layer normalisations, do not need to be altered. Note that this diagram only showed the differences introduced by NaViT and other parts of the model architecture is identical to the Vision Transformer [1].

We demonstrate that: (i) Randomly sampling resolutions at training time significantly reduces training cost. (ii) NaViT results in high performance across a wide range of resolutions, enabling smooth cost-performance trade-off at inference time, and can be adapted with less cost to new tasks, (iii) Fixed batch shapes enabled by example packing lead to new research ideas, such as aspect-ratio preserving resolution-sampling, variable token dropping rates, and adaptive computation.

These observations have major practical implications. At a fixed computational budget NaViT consistently outperforms ViT. For instance, we match the performance of the top-performing ViT with 4× less compute (Figure 1, left). We identify the substantial increase in the number of training examples processed within the allocated compute budget as the primary contributor to the improved performance over ViT (Appendix B.5) —example packing coupled with variable resolution inputs and variable token dropping enable NaViT-L/16 to process five times more images during training (Table 2). This improved efficiency extends to the fine-tuning process (Figure 1, middle). Furthermore, by exposing NaViT to multiple resolutions during both pre-training and fine-tuning, a single model demonstrates excellent performance when evaluated on various resolutions, significantly advantaging NaViT in terms of inference cost (Figure 1, right).

NaViT's training and adaptation efficiency, and flexible inference, presents a promising avenue for Vision Transformers. Patch n' Pack empowers computer vision systems to transcend limitations imposed by current data and modeling pipelines, enabling ideas that were previously restricted by the constraints of fixed batch shapes, unlocking new possibilities for innovation and advancement.

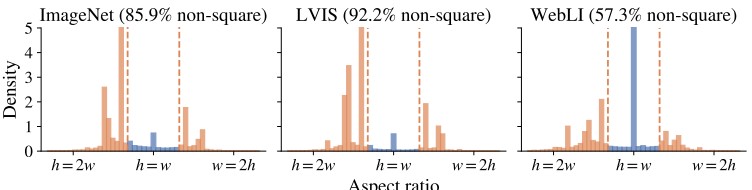

**Figure 3:** Height:width ratios of different datasets; most images are not square-ish (> 20% deviation).

## 2 Method

Deep neural networks are typically trained and run with batches of inputs. For efficient processing on the current hardware this implies fixed batch shapes which in turn imply fixed image sizes for computer vision applications. This coupled with architectural limitations historically associated with convolutional neural networks led to a practice of either resizing or padding images to a fixed size. Both of these have been shown to be flawed: the former harms performance and the latter is inefficient [3]. An analysis of aspect ratios in ImageNet [4], LVIS [5] and WebLI [6] as representative examples of classification, detection and web image datasets, respectively, shows that most images are typilally not square (Figure 3).

In language modelling, it is common to bypass limitations of fixed sequence lengths via *example packing*: tokens from multiple distinct examples are combined in one sequence, which can significantly accelerate training of language models [7]. By treating images as sequences of patches (tokens), we show that Vision Transformers [1] can benefit from the same paradigm, which we call Patch n' Pack. Using this technique ViTs can be trained on images at their "native" resolution, and we name this approach NaViT.

### 2.1 Architectural changes

NaViT is built upon the original ViT, but in principle can use any ViT variant operating on a sequence of patches. To enable Patch n' Pack, we make the following architectural modifications.

**Masked self attention and masked pooling.** To prevent examples attending to each other, additional self-attention masks are introduced. Similarly, masked pooling on top of encoder aims to pool the token representations within each example, resulting in a single vector representation per example in the sequence. Figure 2 presents how the receptive filed of attention is controlled via masking.

**Factorized & fractional positional embeddings.** To handle arbitrary resolutions and aspect ratios, we revisit the position embeddings. Given square images of resolution $R \times R$, a vanilla ViT with patch size $P$ learns 1-D positional embeddings of length $(R/P)^2$ [1]. Linearly interpolating these embeddings is necessary to train or evaluate at higher resolution $R$.

Pix2struct [3] introduces learned 2D absolute positional embeddings, whereby positional embeddings of size [maxLen,maxLen] are learned, and indexed with $(x,y)$ coordinates of each patch. This enables variable aspect ratios, with resolutions of up to $R = P \cdot$ maxLen. However, every combination of $(x,y)$ coordinates must be seen during training.

To support variable aspect ratios and readily extrapolate to unseen resolutions, we introduce factorized positional embeddings, where we decompose into separate embeddings $\phi_x$ and $\phi_y$ of $x$ and $y$ coordinates. These are then summed together (alternative combination strategies explored in Section 3.4). We consider two schema: absolute embeddings, where $\phi(p) : [0, \text{maxLen}] \to \mathbb{R}^D$ is a function of the absolute patch index, and fractional embeddings, where $\phi(r) : [0,1] \to \mathbb{R}^D$ is a function of $r = p/\textit{side-length}$, that is, the relative distance along the image. The latter provides positional embedding parameters independent of the image size, but partially obfuscates the original aspect ratio, which is then only implicit in the number of patches. We consider simple learned embeddings $\phi$, sinusoidal embeddings, and the learned Fourier positional embedding used by NeRF [8].

### 2.2 Training changes

Patch n' pack enables new techniques to be used during training of NaViT.

**Continuous Token dropping.** Token dropping (random omission of input patches during training) [9, 10] has been developed to accelerate training. However, typically the same proportion of tokens are dropped from all examples; packing enables continuous token dropping, whereby the token dropping rate can be varied per-image. This enables the benefits of faster throughput enabled by dropping while still seeing some complete images, reducing the train/inference discrepancy. Further, with packing, the

drop-distribution can vary throughout training, following some pre-defined schedule. In Section 3.3, we explore different schedules and the benefits of flexible token dropping.

**Resolution sampling.** NaViT can be trained using the original resolution of each image. Alternatively, the total number of pixels can be resampled while preserving aspect ratio. In vanilla ViT, there is a tension between greater throughput (training on smaller images), and greater performance (training on larger images, to enable high-resolution at evaluation time). Oftentimes, models are pre-trained at a smaller resolution and finetuned at a higher one [11]. NaViT is much more flexible; it allows mixed-resolution training by sampling from a distribution of image sizes, while retaining each images' original aspect ratio. This allows both higher throughput and exposure to large images, yielding substantial improved performance over equivalent ViTs (in terms of models size and training duration). Section 3.2 explores different sampling strategies, and variable resolution training for pre-training and finetuning.

## 2.3 Efficiency of NaViT

Here we discuss some implications of Patch n' Pack on the computational efficiency of NaViT.

**Self attention cost.** The $\mathcal{O}(n^2)$ cost of attention is a natural concern when packing multiple images into longer sequences. Though many works aim to remove this quadratic scaling [12, 13], we demonstrate here that as the transformer hidden dimension is scaled, the attention becomes an increasingly smaller proportion of the the overall cost, which encompasses the computation cost of the MLP as well. Figure 4 illustrates this trend, indicating a corresponding reduction in the overhead associated with packing examples. In addition to speed considerations, the memory cost of self-attention can pose a challenge for extremely long sequences. However, this challenge can be also addressed by employing memory-efficient methods [14, 15].

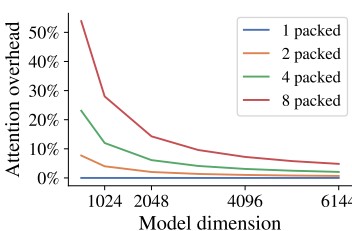

**Figure 4:** Overhead from extra attention due to packing, assuming 256 tokens per image; it diminishes with model scale.

**Packing, and sequence-level padding.** The final sequence lengths containing multiple examples must be fixed. We use a greedy packing approach discussed in Appendix B.3; there typically is no perfect combination of examples exactly adding up to the fixed length and padding tokens have to be used. One could for example dynamically choose the resolution or token dropping rate of the final example in a sequence to exactly fit the remaining tokens; however, we find typically less 2% of tokens are padding tokens, and thus the simple approach is sufficient.

**Padding *examples* and the contrastive loss.** Per-token losses are straightforward to implement with packed sequences. However, many computer vision models are trained with example-level losses, typically applied to a pooled representation. First, this requires modifications to the typical pooling heads to account for packing. Second, multiple pooled representations must be extracted from each sequence. Fixed batch shapes requires an assumption that, from a batch of $B$ sequences, we extract at most $B \times E_{\texttt{max}}$ pooled representations (i.e. $E_{\texttt{max}}$ examples per sequence). If a sequence contains more than $E_{\texttt{max}}$ images, the extra images will be dropped, wasting computation of the model's encoder. If a sequence has less than $E_{\texttt{max}}$ examples, then the loss will process lots of fake padding representations.

The latter is an issue for contrastive learning, where loss computation scales in time and memory $\sim \mathcal{O}(n^2)$. To avoid this, we used the chunked contrastive loss [16], which circumvents the need to gather all data points for the softmax by performing computations on local device subsets and efficiently accumulating the necessary statistics for global softmax normalization. This enable high values of $E_{\texttt{max}}$ (and thus efficient use of the model encoder), without being bottlenecked by the loss.

## 3 Experiments

The base architecture we use for NaViT follows vanilla ViT [1], with the changes to enable packing, described in Section 2.1. In addition, we include small ViT improvements from previous works: query-key normalization and the omission of biases [17], and attention pooling [18].

We pre-train NaViT in two setups: classification training on JFT-4B [18] and contrastive language-image training [19] on WebLI [6]. Typically, for JFT, inception crop is applied pre-training [20, 1], and in both cases, images are resized to a square (distorting aspect ratio). Unless otherwise specified,

all NaViT models are pre-trained without these operations, and preserve aspect ratio. NaViT is implemented in JAX [21] using the FLAX library [22] and built within Scenic [23].

**Classification pretraining.** We pre-train NaViT with supervised classification objective, using a sigmoid cross-entropy loss, following the setup of [17] on JFT-4B [18]. Visual representations are evaluated following the linear evaluation protocol used for ViT [1], where 10 examples per class are used to train a linear classifier on top of frozen representations.

**Contrastive pre-training.** Alongside the image model, we train a text encoder with the same architectural modifications using the contrastive image-text loss [19, 24] (details in Appendix B.2). Packing also provides efficiency improvements on the text-tower, as text sequences do not need to be padded to a fixed lengths, which is the normal setup. The contrastive models are evaluated on zero-shot ImageNet classification and COCO image-text retrieval.

Note that throughout all the pre-training experiments and the transfer to downstream experiments, both the NaViT and baseline models are trained with compute-matched setup. This implies that during pretraining, the same amount of TPU time is used and for downstream tasks, the models are evaluated at the same effective resolution (i.e., the same number of tokens). To be precise, all downstream experiments utilized the top-rightmost points in Figure 1 (ViT-L/16 and NaViT-L/16).

## 3.1 Improved training efficiency and performance

Figure 1 illustrates the JFT pretraining performance of different NaViT models compared to compute-matched ViT baselines [25]. The experimental setup details are provided in Appendix B.1. NaViT consistently surpasses ViT in performance while using the same computational budget across different compute and parameter scales; for example, the performance of the top-performing ViT can be matched by a NaViT with four times less compute. Conversely, the computationally lightest NaViT in Figure 1 is five times more cost-effective than its equivalent ViT counterpart.

The NaViT models benefit from preserved aspect ratios and the ability to evaluate over many resolutions, but the chief contributor here is the significant increase in the number of training examples processed by NaViT within the allocated compute budget. This is achieved through the combination of sampling multiple variable-resolution examples and token dropping, leading to variable size images that are efficiently packed into a similar sequence length as the original model. We ablate these factors below.

## 3.2 Benefits of variable resolution

Here, we deep-dive the benefits of mixed-resolution training. Since we preserve the native aspect ratio, when we refer to "resolution" for NaViT, we mean "effective resolution". That is, images with the same area as a square image with a given resolution. For example, for NaViT a resolution of "128" has the same area of a square 128 x 128 image, but could be 64 x 256, or 170 x 96, etc., and thus has the same inference cost as regular ViT on 128 x 128 images.

**Variable-resolution pre-training.** Lower resolution images require fewer FLOPs to process and hence small resolutions (like 224) are used with fixed-resolution training. With fixed-resolution training, there is a trade-off between throughput and ability to process details and high-resolution images. With NaViT we can mix lower resolution images with large ones to get the best of both worlds.

Figure 5 shows a comparison between two NaViT variants trained at several different resolutions. Here, *all trained for the same number of FLOPs*. (1) Native aspect ratio, but fixed resolution $R = R_{\texttt{max}}$ for different chosen values of $R_{\texttt{max}}$. (2) Variable resolution, where the resolution is distributed as $R \sim \mathcal{U}(64, R_{\texttt{max}})$. Variable resolution models outperform models trained at only that resolution. Even in the best case for fixed resolution, where the train and evaluation resolutions are identical, variable resolution matches or outperforms fixed.

**Variable-resolution finetuning.** Prior works increase resolution late in pre-training or during finetuning, producing higher quality but more expensive models [1, 11]. We finetune NaViT and ViT models at different fixed resolutions, and additionally NaViT at variable resolutions. Figure 6 shows the results of fine-tuning pretrained ViT and NaViT on ImageNet-1k dataset. Performance gains during pretraining transfer well at all resolutions, but two phenomena are particularly interesting: First, NaViT finetuned with variable resolutions ("NaViT 64:512") is as good as a NaViT finetuned at a single resolution (and much better than single-resolution ViT), removing the need to pick a single downstream finetuning resolution. Second, NaViT finetuned at low resolution (64) still obtains good performance when

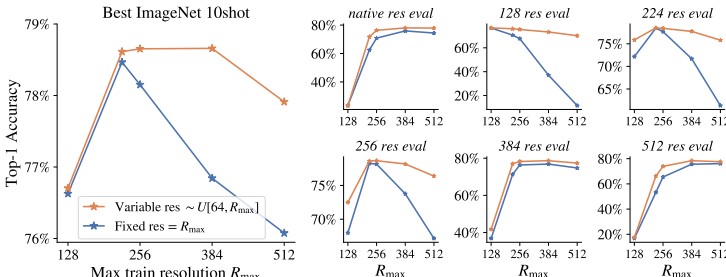

**Figure 5:** At fixed computational cost, sampling lower resolutions increases throughput, improves performance and enables better use of models at varied resolutions. NaViT-B/16 models trained with variable vs. fixed resolutions demonstrate the benefit of mixed resolution.

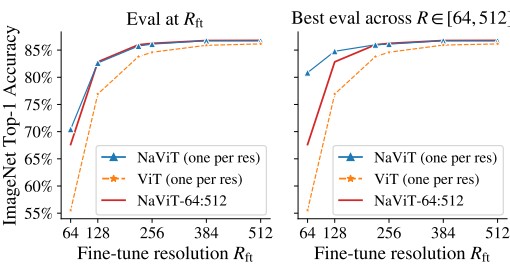

**Figure 6:** Variable-resolution finetuning, JFT B/16 models finetuned on ImageNet at various resolutions. Overall NaViT in all settings (blue, red), outperforms ViT (orange) **Left**: A single NaViT finetuned with variable resolutions (red) is as good as models tuned on only one resolution (blue). **Right**: Mixed-resolution pretraining performs well at high resolution when finetuning at low resolution (left-hand end of blue curve).

evaluated at higher resolutions (Figure 6, right), enabling cheaper adaptation. This ability to perform cheap adaptation of a flexible pre-trained model corroborates findings in [2].

**Resolution sampling strategies.** Packing examples enables diverse resolution sampling strategies. We first consider whether to sample the target *side length* (average height/width, $R$), or the target *area* (i.e. sequence length $\propto R^2$). Sampling the side length from a uniform distribution biases towards lower sequence lengths, whereas sampling the area from a uniform distribution biases towards higher side lengths.

For each image, we sample $u \sim \mathcal{D}$, where $\mathcal{D}$ is a distribution with support $[-1,1]$. We rescale $u$ linearly to $[64,384]$ for sampling side-lengths or $[64^2,384^2]$ for sampling areas. We consider four distributions $\mathcal{D}$: uniform $u \sim \mathcal{U}(-1,1)$, truncated (to $[-1,1]$) standard Normal $u \sim \mathcal{N}_t(0,1)$, and then two other Normals which bias towards lower resolution $u \sim \mathcal{N}_t(-0.5,1)$ and higher resolution $u \sim \mathcal{N}_t(0.5,1)$. The results are shown in Figure 7. Here, the best resolution resampling strategy consistently performs over the default resolution. It is consistently better to

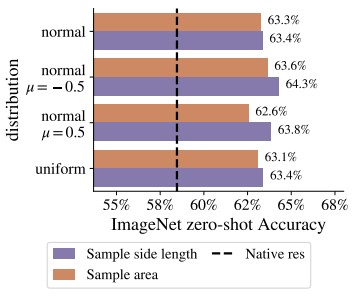

**Figure 7:** Sampling side lengths directly with a bias towards lower resolutions gives overall best performance at a fixed computational budget.

sample side-lengths (orange) as opposed to area (blue), and the best distribution is the truncated normal biasing towards lower values; both of these increase throughput by preferentially sampling smaller sequences.

### 3.3 Benefits of variable token dropping

**Token dropping strategies.**

We experimented with continuously sampled token dropping rates, and with resolution-dependent token dropping rates; both are explained in Appendix B.6.

Fig. 9a compares variable drop rates sampled from a Beta distribution to a constant drop rate, demonstrating consistent improvements from the former. Fig. 9b shows the use of a resolution dependent token dropping rate for models trained with $R \sim \mathcal{U}(64, 384)$ and dropping rates scaled between $[0.5 - \delta, 0.5 + \delta] \propto R$, which further improves over the beta distribution.

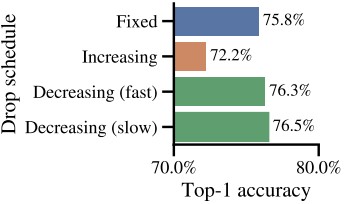

**Figure 8:** Time-varying token dropping rates improves performance and are easily done with Patch n' Pack.

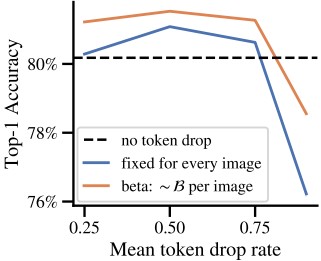 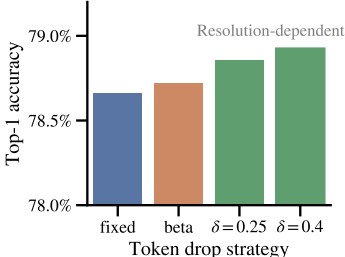

**Figure 9:** Continuous token dropping strategies enabled by sequence packing improves performance

**(a)** Constant vs. Beta-distributed token dropping rates.

**(b)** Resolution dependent token dropping sampled $\in 0.5 \pm \delta$

**Scheduled token dropping rates.** Packing also enables easy variation the token dropping rate during training. By changing the token dropping rate we can better tune the trade-off between number of images seen and information used per image, to maximize the final accuracy while keeping the total training cost constant. We varied the token dropping rate as a function of the number of images seen (details of the schedule in Appendix B.7). Fig. 8 demonstrates that further improvements are possible by reducing the token dropping rate during JFT pretraining of NaViT-B/16.

## 3.4 Positional embeddings

We evaluate our factorized embeddings introduced in Section 2.1, and their design choices. We are interested in both absolute performance, and extrapolation to resolutions outside the training regime. To test this, we train NaViT-B/16 models for 200k steps on JFT, with resolutions $R \sim \mathcal{U}(160, 352)$. We evaluate performance at a range of resolutions, without modification of the embedding variables. We compare to a ViT-B/16 trained at fixed resolution 256 for the same amount of images seen, evaluated at new resolutions using standard interpolation of positional embeddings.

Figure 10a illustrates the disparity among different positional embedding methods when the model is evaluated under the "in-distribution" setup, where the resolutions of training and testing data are within a comparable range. Conversely, Figure 10b shows the model performance across out-of-distribution resolutions, revealing notable distinctions between the approaches.

First, it is clear that the factorized approaches outperform both the baseline ViT and the Learned 2D embeddings from Pix2struct. The latter in particular struggles to generalize to higher resolution, likely because this requires an increasingly long tail of unseen $(x,y)$ pairs. Factorized embeddings are best combined additively (as opposed to stacking or multiplying).

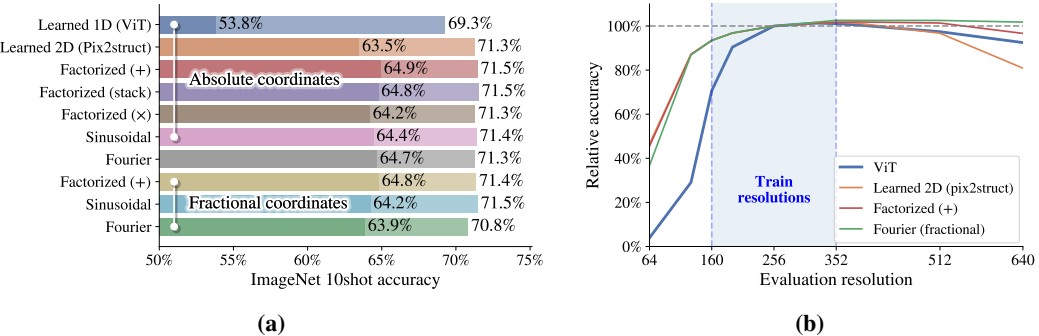

**Figure 10:** Factorized position embeddings improve generalization to new resolutions and aspect ratios. (a) Best (faded) and average accuracies (dark) across resolutions. (b) Accuracy normalized w.r.t. resolution 256.

## 3.5 Other aspects of NaViT's performance

**Out of distribution generalization.** We directly evaluate JFT-pretrained NaViT on downstream datasets, employing a label-map [26] from JFT-4B to ImageNet [4] and robustness-variants (Object-Net [27] and ImageNet-A [28]). We compare the performance to a compute-matched ViT baseline.

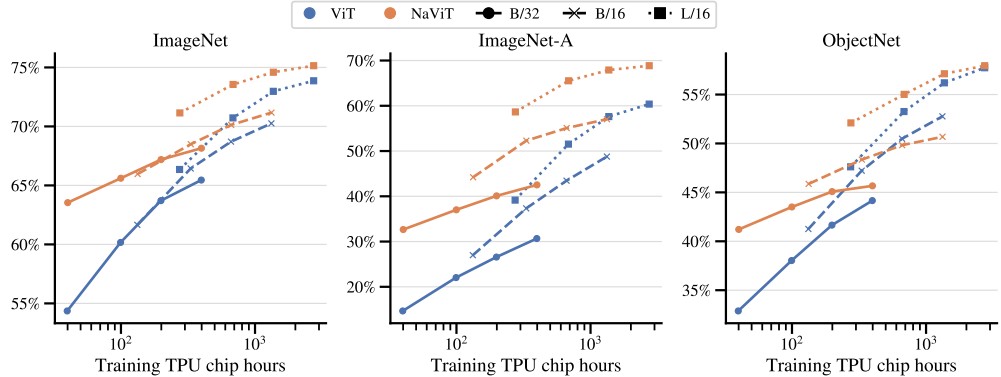

**Figure 11:** Out of distribution evaluation of ViT and NaViT models that were matched for training compute. In addition to improved performance due to more images seen (see also Figure 1), NaViT performs much better on ImageNet-A that has many images with an extreme aspect ratio and important information outside the center crop (Appendix G). Same data as in Table 5.

Figure 11 shows that NaViT compares favorably both on ImageNet as well as datasets variants that were specifically designed to test out of distribution performance. It is interesting to note that NaViT performs much better on ImageNet-A, but ViT catches up on ObjectNet, even though both these datasets contain images that have extreme aspect ratios. We believe this is due to the aspect-preserving center crop that we apply to images for ImageNet-A and ObjectNet classification (same as in [17]), which is a useful prior for ObjectNet, but less so for ImageNet-A (see Appendix G for details). If no crop is applied to images and they're instead simply resized to the image resolution expected by the model (i.e., square for ViT, and aspect preserving resize to the same number of tokens for NaViT), then the observed difference is much larger (see Figure 21 in appendix).

**Calibration.** In addition to the in-depth accuracy analysis, we have also quantfied the quality of the uncertainty computed by the model. In particular, for our ImageNet1K-finetuned models, we computed the expected calibration error [29] of the top prediction, as we vary the number of patches we assign per examples. We find that the calibration error remains *very stable* in the interval $(0.045, 0.047)$ as we vary the number of patches per image in the range $[128, 1024]$, without any post-hoc recalibration. We provide further details in the appendix Appendix F.

**Inference trade-offs.** Given the flexibility of the model, there are several viable approaches how one can maximize the aggregate accuracy under a given compute budget, which we quantify in terms of the latency measured on a Cloud TPUv3 chip. In an online inference setting, choosing an inference strategy translates to an algorithm how to allocate a fixed number of tokens per example. As we show in Fig. 1, NaViT offers much better trade-offs than ViT and it shows strong diminishing returns, s.t., even relatively few patches provide highly competitive results. In Appendix D we further study a cascading approach, which both provides Pareto optimal models and gives more precise trade-off opportunities.

**Fairness signal annotation.** We investigate annotating images with fairness-related signals, such as those pertaining to gender and ethnicity. Prior research has shown that metrics, such as group calibration, are susceptible to labeling inaccuracy, particularly for underrepresented groups. In addition, this problem persists even when accounting for label noise during training [30]. Thus, reducing the labeling error of fairness signals improves the reliability of bias mitigation and post-hoc auditing [31]. To explore whether NaViT can help in overcoming these challenges, we train annotators on FairFace [32] and CelebA [33] datasets as linear probes (i.e. using frozen features produced by NaViT or ViT), before comparing their accuracy.

First, NaViT provides representations of higher quality that improve the accuracy of fairness signal annotation, even when dealing with square images. Original images are of size $448 \times 448$ in FairFace and $178 \times 218$ in CelebA, and we resize them to area $224^2$ while preserving aspect ratios in NaViT. Despite having the same sequence length, NaViT provides a higher prediction accuracy than ViT, as shown in Figure 12 (left). We verify statistical significance using the Wilcoxon signed-rank test test [34], which reveals that the improvement in NaViT is significant with $p = 3 \times 10^{-4}$.

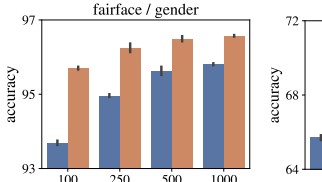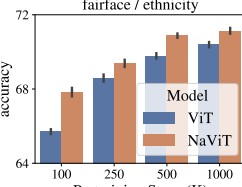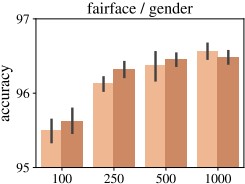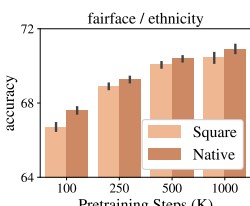

**Figure 12:** Evaluating the accuracy of annotators trained on fairness-related signals using either NaViT-L/16 or ViT-L/16: **Left**: NaViT offers better representations that improve the accuracy of annotators. **Right**: Using native aspect ratios in NaViT results in a higher performance when compared to resizing images to squares.

Second, we apply inception-style cropping [35] with a fixed minimum area of 50%. This changes the aspect ratio but maintains native resolution. Similar to before, we resize all cropped images to area $224 \times 224$, either as square images or with native aspect ratios. Figure 12 (right) shows that native aspect ratios in NaViT improve accuracy, which is statistically significant at the 95% confidence level ($p = 0.02$). Appendix H contains the full set of figures for other attributes in CelebA and Fairface.

## 3.6 Other downstream tasks

**Semantic segmentation.**

We finetune NaViT to semantic segmentation on ADE20k dataset [36], following the linear decoder protocol of Segmenter [37]. We use ViT-L/16 as baseline and compare its performance with that of NaViT-L/16. Both models are pre-trained on JFT-4B [18] with comparable compute budget.

We experiment with different maximum resolution $R_{max}$ at finetuning time: ViT takes in random square crops of resolution $R_{max} \times R_{max}$ while NaViT's inputs are randomly resized (with preserved aspect ratio) so that the total number of pixels is $R_{max}^2$. This way, we have the same finetuning cost for the ViT and NaViT models. Note that following common practice, in order not to alter the ground truth segmentation maps, both models are evaluated at the native resolution [37, 38, 36]. This means that for ViT, the square predictions are resized from square back to native resolution. We observe in Figure 13 that NaViT outperforms ViT when transferred to semantic segmentation with the same maximum finetuning resolution $R_{max}$. Note that NaViT at $R_{384}$ beats ViT at $R_{512}$ while being twice as fast (see Appendix D). An advantage of NaViT over ViT is that it benefits from flexibility of resolution during training and can ingest seamlessly square and non-square images.

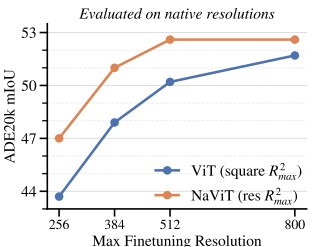

**Figure 13:** NaViT transfers competitively to semantic segmentation. We transfer ViT-L/16 and NaViT-L/16 on ADE20k with fine-tuning at different resolutions. ViT consumes square images while NaViT preserves the aspect ratio of the original images, while maintaning the total number of pixels the same as ViT.

**Object detection.**

Native-resolution training may be especially beneficial for fine-grained tasks such as object detection, which require image understanding at many different spatial scales. We use compute matched NaViT and ViT models as backbones for OWL-ViT-L/14 object detectors [39], following the OWL-ViT protocol for training and evaluation. Results are presented in Table 1. The NaViT-based detector

**Table 1:** NaViT improvements carry over to object detection.

|  | ViT-L/14 | NaViT-L/14 |
| --- | --- | --- |
| ImageNet zeroshot | 68.3% | 72.9% |
| LVIS AP | 23.3% | 28.3% |
| LVIS AP rare | 17.2% | 24.3% |

performs significantly better on both common and unseen LVIS "rare" classes. These experiments used shorter pre-training than the original OWL-ViT and thus reach lower absolute performance; the relative difference nonetheless suggests that NaViT produces strong representations for fine-grained vision tasks.

**Video Classification.** Video processing with transformers is inherently challenging due to the heterogeneity of the underlying spatio-temporal signals. In practice, cumbersome protocols have to be

established both for training and evaluation (e.g. spatial and temporal multi-cropping) [40]. NaViT alleviates some of these challenges by not only allowing training over different resolutions, but also over different temporal durations. We fine-tuned NaViT trained on JFT for Kinetics400 [41] classification by extracting three different spatio-temporal patches "tubelets"'. [42], extending the positional embedding to include the temporal dimension and initializing the embedding kernel using "central frame embedding" [40]. We posit that NaViT is an excellent starting point for multi-scale training due to the resolution diversity at training time. We observe that NaViT-L achieves competitive performance with ViViT-L (80.4%) in approximately 6x less epochs, without multi-crop evaluation. Note that the Kinetics400 dataset used here contains less data than prior works [40].

## 4 Related work

**Flexible Vision Transformers.** FlexiViT [2] developed a novel kernel resizing approach which enables variable "resolution" via models which support multiple patch sizes. This is viewed as unifying multiple distinct models, and they study the relationship with distillation and neural architecture search. Pix2struct [3] supported variable aspect ratios with a novel positional embedding schema, and demonstrated significant efficiency and performance improvements for non-natural imagery such as chart and document understanding.

**Multiscale Vision Transformers.** Using feature maps at multiple spatial scales is a common approach to localization tasks such as segmentation and detection [43]. Many works developed Vision Transformers which do the same [44, 45], though some dispute the necessity for simple localization tasks [46]. NaViT does not build hierarchical representations using multiple scales; we believe combining the unique flexibility of our approach with the benefits of this modelling family is a promising avenue.

**Accelerating training with mixed resolutions.** Image modelling works considering resolution largely focus on accelerating pretraining with a fixed, low resolution [47, 48]. FixRes [11] is a popular technique whereby resolution is increased in a final stage of pretraining, and has been used in many subsequent works [19, 6]. PaLI [6], for example, successively increases the resolution of the vision backbone during its generative training. This approach's main downside is its irreversibility: compute cannot be scaled back by reducing resolution after tuning at the higher and more expensive resolution.

Multigrid training [49] is the most closely related work. The main idea is accelerate video modelling by processing large batches with "coarse" spatiotemporal resolution early in the training, followed by a "finer" resolution later. To this end, the authors apply a hierarchical grid sampling schedule coupled with appropriate learning rate scaling. In contrast, Patch n' Pack enables effortless incorporation of mixed resolutions without complex schedules or training pipelines.

**Token dropping for improved efficiency.** Research initially explored random token dropping [9, 10, 50]. Follow ups demonstrated benefits from considering structured [51] or "importance" based strategies [52, 53]. Better strategies will likely further boost performance, and Patch n' Pack sidesteps the fixed minibatch shape restriction which limited these works.

## 5 Conclusions and future work

We have demonstrated that Patch n' Pack—the simple application of sequence packing to vision transformers—significantly improves training efficiency. The resultant NaViT models can be applied to many resolutions at inference time, and cheaply adapted to new tasks. We discuss limitations and impacts in Appendix A, but overall Patch n' Pack enables a wide variety of research previously hindered by the need for fixed batch shapes, including adaptive computation and new algorithms for improving training and inference efficiency.

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

# A Broader impacts & limitations

**Broader Impacts**

NaViT enables training of vision transformers on variable size inputs, which has a profound impact on advancing adaptive computation research. By training models to handle various input size, we can explore adaptive computation techniques that dynamically adjust the computational resources based on the specific requirements of a given input. This flexibility opens up new avenues for implementing ideas that aim at adjusting allocation of compute and improving efficiency in vision tasks per input. Furthermore, NaViT computational efficiency unlocks the potential for scaling up pre-training of vision models. With the ability to handle different resolutions, models can effectively tackle more complex and diverse visual data, allowing for the development of larger and more powerful vision models.

**Limitations**

There is a wide range of applications that could benefit from a model capable of processing inputs of different resolutions, including OCR or document understanding with the use of vision models. Although we highlight the considerable advantages of employing NaViT and provide a comprehensive analysis of common computer vision tasks, we did not specifically investigate the benefits of NaViT in these particular applications. We consider this area as a priority for future research and follow-up work.

**Implementation complexities** The modelling changes shown in Figure 1 can complicate ViT; in particular, packing-aware changes to the loss function may be complex for more complicated usecases.

**Computational overhead** Done carelessly, significant extra cost can be introduced due to sequence packing (Discussed in Figure 4). NaViT can however be run with the same sequence length as a corresponding ViT model (e.g. ViT at resolution 384 with sequence length 576 vs NaViT at resolutions 64:384 with sequence length 576; the latter will fit on average $\sim 2\times$ as many images).

**Data augmentation aspects** were not explored in this work; firstly, it is possible that token dropping and variable resolution can also act as data augmentations, which was not exhaustively studied here. Secondly, common data augmentations such as mixup and various random cropping schema assume square, constant resolution images, and analogues supporting variable aspect ratio and size were not developed here.

**Scale**; experiments were performed in a large-dataset regime, and future work should validate these results at smaller scale. This however necessitates research into data augmentation techniques as mentioned above.

# B Training details

## B.1 Classification pretraining

The experiments and ablations in the paper are with ViT-B/32, ViT-B/16, and ViT-L/16. We use a reciprocal square-root learning rate schedule, with linear warmup and cooldown, with a maximum value of $8e-4$, and phases. We follow [18, 54] and use a higher weight decay of 3.0 on the head compared to the body's weight decay of 0.03 during upstream training to improve transfer to downstream tasks. For our experiments, we evaluated both NaViT and ViT models using configurations B/32, B/16, and L/16. Each ViT model was trained with varying compute budgets, with cooling down at different stages of training. We trained a corresponding NaViT model for each ViT size and computational budget, allowing us to perform "compute-matched" comparisons [25].

Table 2 presents the pretraining specifications for both ViT and NaViT models. During the pretraining phase, ViT models were trained using images of size $224\times224$. In contrast, NaViT models uniformly sampled a value, denoted as $r$, between 64 and 256 and resized the image to have a total of $r^2$ pixels while preserving the aspect ratio. Although training NaViT models on native resolutions is possible, we empirically discovered that sampling a resolution provides greater control over maximizing the number of examples observed during pretaining within a fixed computational budget while maintaining performance across different resolutions in downstream tasks. Additionally, by controlling the resolution, we can ensure efficient packing by tuning the sequence length and limit padding to less than 2%.

**Table 2:** Pre-training details of ViT and NaViT with supervised classification.

| Name | TPU Hours | Train Steps | Cooldown Steps | Sequence Length | Images Per Seq. | Batch Size | Training Images |
|---|---|---|---|---|---|---|---|
| ViT-B/32 | $7.9\times10^1$ | $1.0\times10^5$ | $1.0\times10^4$ | 49 | 1.0 | $\approx4.0\times10^3$ | $4.0\times10^8$ |
| | $1.9\times10^2$ | $2.5\times10^5$ | $5.0\times10^4$ | 49 | 1.0 | $\approx4.0\times10^3$ | $1.0\times10^9$ |
| | $3.9\times10^2$ | $5.0\times10^5$ | $1.0\times10^5$ | 49 | 1.0 | $\approx4.0\times10^3$ | $2.0\times10^9$ |
| | $7.9\times10^2$ | $1.0\times10^6$ | $1.0\times10^5$ | 49 | 1.0 | $\approx4.0\times10^3$ | $4.0\times10^9$ |
| ViT-B/16 | $2.6\times10^2$ | $1.0\times10^5$ | $1.0\times10^4$ | 196 | 1.0 | $\approx4.0\times10^3$ | $4.0\times10^8$ |
| | $6.6\times10^2$ | $2.5\times10^5$ | $5.0\times10^4$ | 196 | 1.0 | $\approx4.0\times10^3$ | $1.0\times10^9$ |
| | $1.3\times10^3$ | $5.0\times10^5$ | $1.0\times10^5$ | 196 | 1.0 | $\approx4.0\times10^3$ | $2.0\times10^9$ |
| | $2.6\times10^3$ | $1.0\times10^6$ | $1.0\times10^5$ | 196 | 1.0 | $\approx4.0\times10^3$ | $4.0\times10^9$ |
| ViT-L/16 | $5.4\times10^2$ | $1.0\times10^5$ | $1.0\times10^4$ | 196 | 1.0 | $\approx4.0\times10^3$ | $4.0\times10^8$ |
| | $1.3\times10^3$ | $2.5\times10^5$ | $5.0\times10^4$ | 196 | 1.0 | $\approx4.0\times10^3$ | $1.0\times10^9$ |
| | $2.7\times10^3$ | $5.0\times10^5$ | $1.0\times10^5$ | 196 | 1.0 | $\approx4.0\times10^3$ | $2.0\times10^9$ |
| | $5.4\times10^3$ | $1.0\times10^6$ | $1.0\times10^5$ | 196 | 1.0 | $\approx4.0\times10^3$ | $4.0\times10^9$ |
| NaViT-B/32 | $7.9\times10^1$ | $9.8\times10^4$ | $1.0\times10^4$ | 64 | 5.41 | $\approx2.2\times10^4$ | $2.1\times10^9$ |
| | $1.9\times10^2$ | $2.4\times10^5$ | $5.0\times10^4$ | 64 | 5.41 | $\approx2.2\times10^4$ | $5.3\times10^9$ |
| | $3.9\times10^2$ | $4.8\times10^5$ | $1.0\times10^5$ | 64 | 5.41 | $\approx2.2\times10^4$ | $1.0\times10^{10}$ |
| | $7.9\times10^2$ | $9.7\times10^5$ | $1.0\times10^5$ | 64 | 5.41 | $\approx2.2\times10^4$ | $2.1\times10^{10}$ |
| NaViT-B/16 | $2.6\times10^2$ | $9.3\times10^4$ | $1.0\times10^4$ | 256 | 4.87 | $\approx1.9\times10^4$ | $1.8\times10^9$ |
| | $6.6\times10^2$ | $2.3\times10^5$ | $5.0\times10^4$ | 256 | 4.88 | $\approx1.9\times10^4$ | $4.6\times10^9$ |
| | $1.3\times10^3$ | $4.6\times10^5$ | $1.0\times10^5$ | 256 | 4.88 | $\approx1.9\times10^4$ | $9.2\times10^9$ |
| | $2.6\times10^3$ | $9.2\times10^5$ | $1.0\times10^5$ | 256 | 4.88 | $\approx1.9\times10^4$ | $1.8\times10^{10}$ |
| NaViT-L/16 | $5.4\times10^2$ | $9.7\times10^4$ | $1.0\times10^4$ | 256 | 4.88 | $\approx1.9\times10^4$ | $1.9\times10^9$ |
| | $1.3\times10^3$ | $2.4\times10^5$ | $5.0\times10^4$ | 256 | 4.87 | $\approx1.9\times10^4$ | $4.8\times10^9$ |
| | $2.7\times10^3$ | $4.8\times10^5$ | $1.0\times10^5$ | 256 | 4.87 | $\approx1.9\times10^4$ | $9.6\times10^9$ |
| | $5.4\times10^3$ | $9.6\times10^5$ | $1.0\times10^5$ | 256 | 4.88 | $\approx1.9\times10^4$ | $1.9\times10^{10}$ |

## B.2 Contrastive pretraining

We use the 32000 token T5 [55] sentencepiece [56] tokenizer. By default, text sequences are truncated to a maximum length of 24. No token dropping is used for text. Models are trained under the same optimization regime as the classification models, but with a learning rate of $3\times10^{-3}$. Weight decay of $1\times10^{-6}$ is applied consistently to all kernels in the model (no change for projection heads). By default, image side-lengths are sampled $\sim\mathcal{U}(64, R_{\max})$, and no other image augmentations are applied.

## B.3 Packing algorithm

Packing of examples into sequences is done alongside batching. A simple greedy approach is used which adds examples to the first sequence with enough remaining space. Once no more examples can fit, sequences are filled with padding tokens, yielding the fixed sequence lengths needed for batched operations. Such simple packing algorithm can lead to a significant padding, depending on the distribution of length of inputs. There are several methods to address such limitations, like bin packing [7], which allows minimizing the padding. Here, in NaViT, since controlling the resolutions we sample, we can ensure efficient packing by tuning the sequence length and limit padding to less than 2%.

## B.4 Pre-processing and augmentation

For both classification and contrastive pre-training, for the baseline models, we follow the original Vision Transformer paper [1], and follow-ups; models are trained on images with resolution $224\times224$, with inception crop followed by random horizontal flipping. For NaViT, we have no preprocessing besides resizing to the sampled resolution and token dropping.

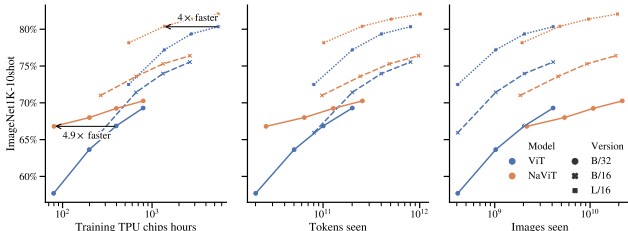

**Figure 14:** Training on more images given a fixed computational budget, is a key factor contributing to the computational efficiency of NaViT compared to to the standard ViT.

## B.5 Sample efficiency with NaViT

Based on the experiments we ran, we believe a lot of the observed performance benefits are due to speeding up training (i.e., seeing more images in a given compute budget). This is evident in Figure 1, where relative performance gains diminish with longer training. We can also show this by plotting the number of images seen on the x-axis:

If compute cost was not a concern, for a given number of images seen, it is best to train on purely large resolution, with no token dropping. Asymptotically we therefore expect the token dropping and mixed resolutions to eventually hurt NaViT in unlimited compute regime. This assumes a large enough dataset that overfitting is not an issue. Eventually one may enter a regime where the token dropping and variable resolution is a useful data augmentation to prevent overfitting, which may alter results.

## B.6 Sampling token dropping rates

**Sampling with a beta distribution** We use a parameterisation based on the mean $d_\mu$ and standard deviation $\sigma$. We aim to sample dropout rate $d \in [0.0, d_{\texttt{max}}]$, with some mean $d_\mu$.

Accordingly, we sample $u \in [0,1] \sim \mathcal{B}(\alpha, \beta)$ and set drop rate $d = u \times d_{\texttt{max}}$. $\alpha$ and $\beta$ are set such that the mean of $u$ is $u_\mu = \frac{d_\mu}{d_{\texttt{max}}}$. The maximum supported variance for a beta distribution of mean $u_\mu$ is $u_\mu(1-u_\mu)$; we pick by default a variance $\sigma^2 = 0.3u_\mu(1-u_\mu)$, which we found to work well in practice. The resultant distributions of token dropouts for different settings of $d_\mu$ and $d_{\texttt{max}}$ are shown in Figure 15a.

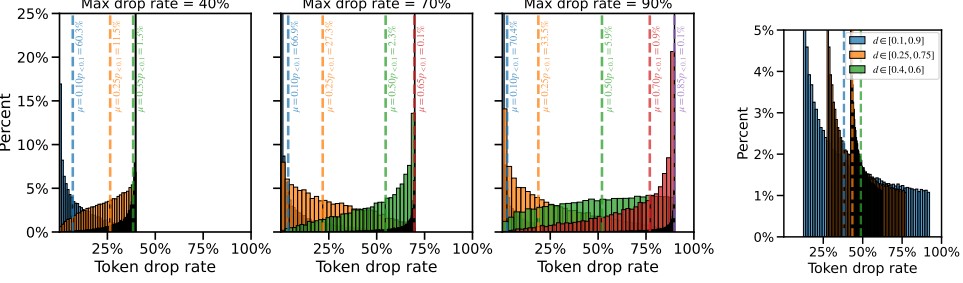

**(a)** Beta-sampled token drop rates parameterised by the mean $\mu$ and the max drop rate $d_{\texttt{max}}$

**(b)** Sampled resolution-dependent token drop rates

**Sampling resolution-dependent dropping rates** Given input data with sequence lengths ranging from $s_{\texttt{min}}$ to $s_{\texttt{max}}$, we sample dropout rate $d$ from a truncated normal distribution $d \sim \mathcal{N}_{\texttt{trunc}}(\mu, 0.02)$, where samples more than two standard deviations away from $\mu$ are rejected.

The mean of this distribution $\mu$ is set according to the minimum and maximum token dropping rates $d_{\texttt{min}}$ and $d_{\texttt{max}}$, and simply scales linearly with the sequence length $s$ (such that $s = s_{\texttt{min}}$ has $\mu = d_{\texttt{min}}$ and $s = s_{\texttt{max}}$ has $\mu = d_{\texttt{max}}$.

Figure 15b shows example distributions of sampled drop rates given inputs with resolution $R \sim \mathcal{U}(64, 384)$, and different values of $d_{\texttt{min}}$ and $d_{\texttt{max}}$.

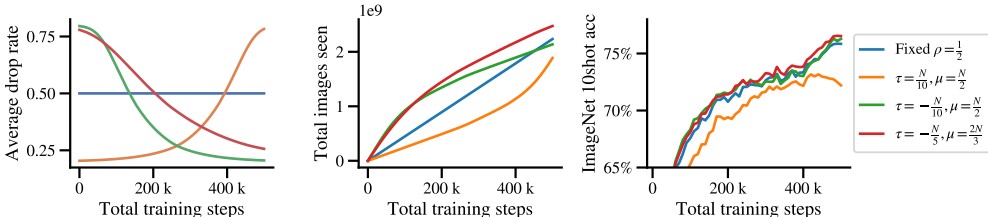

**Figure 16:** Decreasing the token dropping rate along training improves the ImageNet 10shot accuracy using the same pre-training resources. $N$ is the total number of training examples seen with a fixed token dropping rate of $\rho = \frac{1}{2}$.

### B.7   Scheduling token dropping rates

We experiment with a token dropping schedule which varies with total number of images seen. In particular, the rate applied for the $n$-th processed image during training is given by:

$$\rho(n;\rho_{\min},\rho_{\max},\mu,\tau) = \rho_{\min} + (\rho_{\max} - \rho_{\min}) \cdot \sigma\left(\frac{n-\mu}{\tau}\right), \tag{1}$$

where $\sigma$ represents the sigmoid function; $\rho_{\min}$, $\rho_{\max}$ control the minimum and maximum dropping rate applied; and $\mu$ and $\tau$ control the shape of the schedule. We experimented with both increasing ($\tau > 0$) and decreasing ($\tau < 0$) schedules. In all cases we set $\rho_{\min} = 0.2$ and $\rho_{\max} = 0.8$. Figure 16 shows that, by decreasing the dropping rate throughout training, one can improve the final accuracy, at fixed training cost. Conversely, increasing the token dropping rate harms performance.

## C   Model information

### C.1   Positional embeddings

Extending ViTs to variable input sizes necessitates rethinking positional embeddings added to every token after embedding. We considered several variants of positional embeddings, and evaluated them based on (1) the best performance model using them achieve within training distribution of input sizes; and based on (2) how well these models perform when evaluated on image sizes outside of the training distribution. Results and discussion of these experiments can be found in Section 3.4.

Broadly, we considered positional embeddings that varied along three axes: (1) whether they were learned, parametric or fixed; (2) whether they were absolute or fractional; and (3) whether they are factorized.

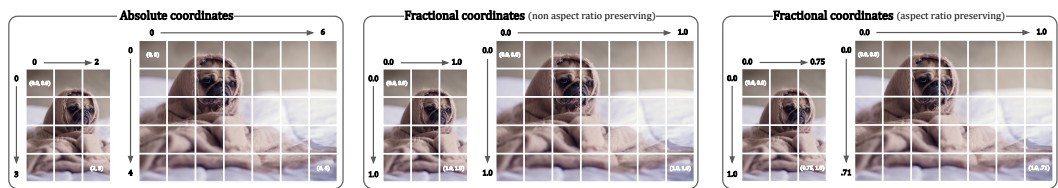

*Image credit: Matthew Henry* `burst.shopify.com/photos/dog-staying-warm`

**Figure 17:** We use two different views of the same image, of resolutions 96×128 and 224×160, and demonstrate different coordinate systems when using patch size 32.

**Absolute and fractional coordinates** A natural way of indexing token within an image is to select *a priori* a maximum possible image side length (shared for width and height) $\mathrm{maxLen}$, and to assign to token integer coordinates $(x,y)$ based on their original location within the image. Embedding coordinates defined in this way allow models to consume images with resolutions up to $R = P \cdot \mathrm{maxLen}$. However, when *learned* absolute coordinate embeddings are considered, extreme values of $x$ and $y$ must also be observed during training, which necessitates training on images with varied aspect ratios and limits models generalisation.

To alleviate the necessity of observing extreme aspect ratios and image size during learning of positional embeddings, we also consider fractional coordinates, which are normalized to the actual size of the input image and are obtained by dividing the absolute coordinates $x$ and $y$ above by the number number of columns and rows respectively, i.e. the corresponding side length. Doing this allows the model to observe extreme token coordinates during training, which intuitively should help with generalization to higher resolutions. However, this is accomplished at the cost of obfuscating the input images aspect ratio.

**Factorized embeddings** We further consider whether coordinates $x$ and $y$ should be embedded independently or jointly. In case of independent embedding, the two coordinates $x$ and $y$ are embedded independently, and their embeddings are combined via addition or by stacking. For joint embeddings and embedding for each position $(x,y)$ is obtained directly.

**Learned, parametric and fixed positional embeddings** Finally, we also explored the relative benefits of fixed, learned and parametric embeddings. For fixed embeddings we followed [57] and used sinusoidal positional embeddings, and learned embeddings were implemented as in [1].

For parametric positional embeddings we followed [8] and used Fourier embeddings. Specifically, coordinates $(x,y)$ were mapped using a single linear layer before applying $\sin$ and $\cos$ activations to them, and stacking the results to obtained the positional embeddings.

**Experiments** Because not all combinations of the above embedding choices are equally promising or natural, we experimented only with subset of them shown in Table 3 and Figure 10a.

**Table 3:** Classification of positional embedding experiments from Figure 10a.

| Name | Coordinates | Type | Factorized |
|---|---|---|---|
| Learned 1D (ViT) | Absolute | Learned | No, position in flatted token sequence |
| Learned 2D (Pix2struct) | Absolute | Learned | No |
| Factorized abs. $(+)$ | Absolute | Learned | Yes, sum |
| Factorized abs. (stack) | Absolute | Learned | Yes, stack |
| Factorized abs. $(\times)$ | Absolute | Learned | Yes, product |
| Fourier abs. | Absolute | Parametric | No |
| Sinusoidal abs. | Absolute | Fixed | Yes, stack |
| Factorized frac. $(+)$ | Fractional | Learned | Yes, sum |
| Fourier frac. | Fractional | Parametric | No |
| Sinusoidal frac. | Fractional | Fixed | Yes, stack |

In sinusoidal and factorised embeddings experiments with fractional coordinates fractional coordinate embeddings were obtained from absolute coordinate embeddings via bilinear interpolation.

## D   Inference strategies

We performed various experiments to measure model quality for given runtime cost. The runtime can be tuned by changing the number of processed patches, or by using choosing different size of the model.

We firstly looked at how model quality changes in respect to decreasing area of the image compared to native resolution, presented in Figure 18a. We observed that on ImageNet [4] model retains most of the quality down to 40% of the image size. After that, the quality drastically decreases. On the other hand, increasing the size of the image have a diminishing return in quality. This can be directly compared with random token dropping as an alternative to resizing, which showed to be very ineffective way to decrease number of patches during inference - Figure 18b.

Please note that this highly depends on the native resolution of the images in the dataset - e.g. dataset with twice as big images than ImageNet can probably be safely resized to 20% of area.

A better way to quantify the performance is by giving a constant compute budget corresponding to number of patches. Figure 19a shows that resizing the image (while preserving aspect ratio) to 256 tokens retains most of the quality (within 0.3%). This corresponds to 256x256 area (given patch size of 16). At 128 tokens (181x181 area) the quality difference reaches 1% and drops significantly after that.

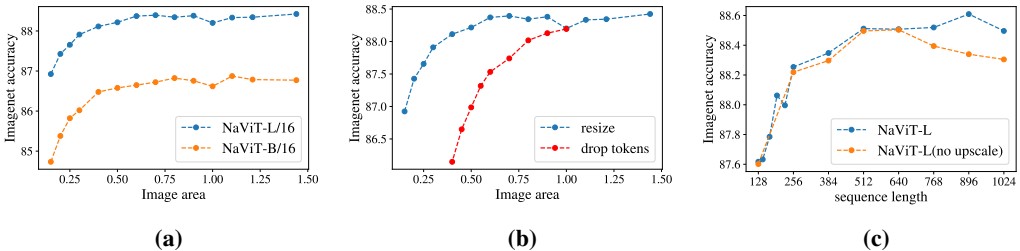

**(a)**        **(b)**        **(c)**

**Figure 18:** (a) The effect of resizing the image. (b) Dropping random tokens is ineffective way to decrease number of patches compared to resizing the image. Data from NaViT-L/16. (c) Given number of patches as compute budget, it is beneficial to upscale the image.

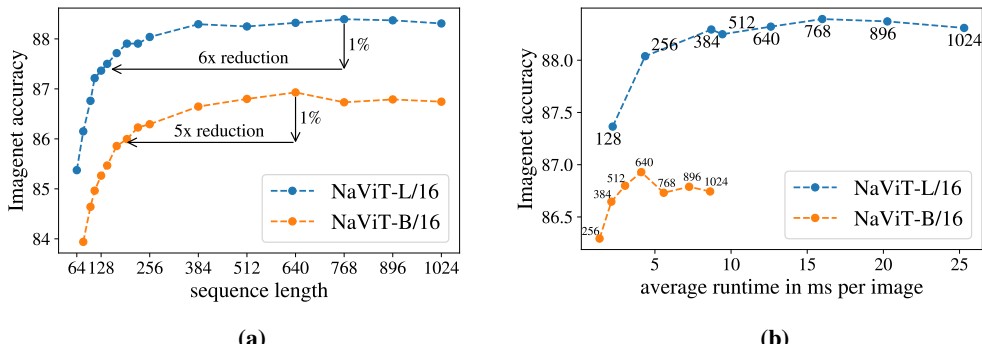

**(a)**        **(b)**

**Figure 19:** (a) Quality on ImageNet in respect to number of patches (sequence length). (b) Runtime of models compared to the accuracy on ImageNet.

Here we also resized the image past its native resolution in case it already fit the given sequence length budget. We observed that it is beneficial to resize the image to the given sequence length past the native resolution to keep monotonic increase in quality, which is showed on Figure 18c.

Figure 19b presents the runtime of NaViT-L/16 and NaViT-B/16 for different sequence lengths. We can see that NaViT-L/16 at sequence length 128 is as fast as NaViT-B/16 with sequence length 512, while having almost 1% difference in quality.

# E    Cascades

Another strategy to be more compute efficient would be to assign more tokens (and thus FLOPs) to the examples deemed hard by the model. This is in particular interesting for bulk inference workloads, where one can amortize over large datasets and where only the total inference time matters.

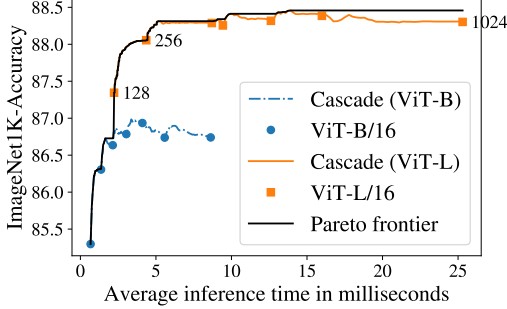

**Figure 20:** Performance of a model cascade versus the average inference time. The labels at the select points denote the number of tokens at that scale.

To evaluate the feasibility of this approach, we consider two sequence lengths $n_1$ and $n_2$ with respective inference times $t_1$ and $t_2$. Then, we (i) send all examples though the model with $n_1$ tokens, and send only the $\alpha \in (0,1)$-fraction deemed hardest (those with the smallest maximum probability) to the model with $n_2$ tokens. To have an input almost exactly fit into $n_1$ or $n_2$ tokens we perform and aspect ratio preserving resize. Hence, the total amortized inference time per-example is $t_i + \alpha t_2$, while the accuracy obtained by combining the accuracy of the first model on the $1 - \alpha$ most-confident fraction of the data, and the performance of the more expensive model on the remaining data. By considering several pairs of models and varying $\alpha$ we obtain the plot in Figure 20. As we can see this strategy is indeed useful and provides not only the best performing models at given compute budgets, but because $\alpha$ is a real parameter one can obtain very fine-grained trade-offs.

## F  Calibration

To evaluate behaviour of the predicted uncertainties with scale, we compute the calibration error of a -B sized ImageNet-finetuned model (the -L model performs similarly). Note that these models were trained with sigmoid loss, i.e., the 1000 labels were predicted independently without enforcing that the probabilities should sum up to 1. As we varied the sequence of tokens per example between 128 and 1024, we obtained very stable calibration errors (top-1, using $\ell_1$ and 30 buckets, i.e., the settings from [29]), which we present in Table 4.

**Table 4:** Expected calibration error on ImageNet-1K with varying sequence lengths.

| Sequence Length | 128 | 256 | 384 | 512 | 640 | 768 | 1024 |
|---|---|---|---|---|---|---|---|
| Calibration Error | 0.047 | 0.046 | 0.048 | 0.047 | 0.047 | 0.046 | 0.045 |

## G  Out of distribution evaluation

For ViT, we apply the "Crop" strategy from [17], namely an aspect-preserving crop of the central 75% of the image for ObjectNet and ImageNet-A, and square resize followed by a 87.5% central crop for the other datasets. We also apply a simple "Resize" strategy that does not crop the images. For NaViT, both the "Crop" and the "Resize" strategy do an aspect preserving resize of the target images.

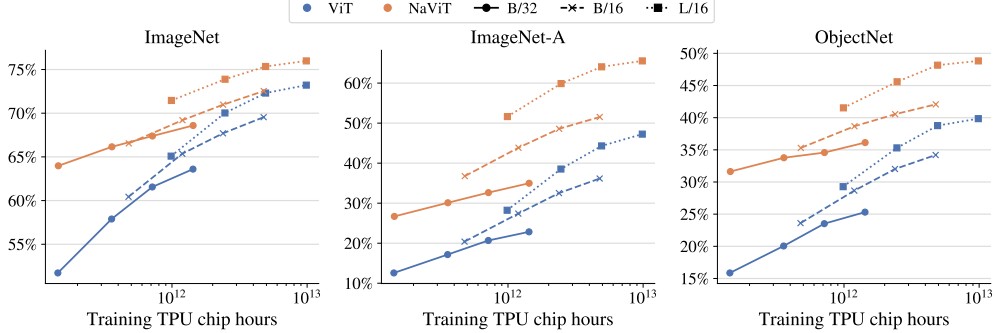

**Figure 21:** Same evaluation as in Figure 11, but without any special preprocessing of the images before the evaluation. Employing a simple resize (square for ViT, aspect preserving for NaViT) results in much better performance on datasets that have images with an extreme aspect ratio. Same data as in table Table 5.

## H  Fairness Signal Annotation

In Figure 22, we demonstrate that using native image resolution improves the performance of fairness signal annotation. Prior research has shown that metrics, such as group calibration, are vulnerable to labeling errors, particularly for underrepresented groups. Moreover, this problem persists even when accounting for label noise during training [30]. Thus, reducing the labeling error of fairness signals has the potential of improving the reliability of bias mitigation and post-hoc auditing [31]. Nevertheless,

**Table 5:** Detailed results of out evaluation of pretrained models with a label-map (see Section 3.5). Same data as in Figure 11 and Figure 21.

| | | | ImageNet | | ImageNet-A | | ObjectNet | |
|---|---|---|---|---|---|---|---|---|
| | | | ViT | NaViT | ViT | NaViT | ViT | NaViT |
| | | Compute | | | | | | |
| Crop | B/32 | $7.9\times10^1$ | 54.4 | 63.5 | 14.7 | 32.7 | 32.9 | 41.2 |
| | | $1.9\times10^2$ | 60.2 | 65.6 | 22.0 | 37.0 | 38.0 | 43.5 |
| | | $3.9\times10^2$ | 63.7 | 67.2 | 26.6 | 40.1 | 41.7 | 45.1 |
| | | $7.9\times10^2$ | 65.5 | 68.2 | 30.7 | 42.5 | 44.2 | 45.7 |
| | B/16 | $2.6\times10^2$ | 66.4 | 68.5 | 37.3 | 52.3 | 47.2 | 48.4 |
| | | $6.6\times10^2$ | 61.7 | 66.0 | 27.0 | 44.2 | 41.3 | 45.9 |
| | | $1.3\times10^3$ | 70.3 | 71.2 | 48.8 | 57.0 | 52.8 | 50.7 |
| | | $2.6\times10^3$ | 68.7 | 70.1 | 43.5 | 55.1 | 50.5 | 49.8 |
| | L/16 | $5.4\times10^2$ | 70.7 | 73.6 | 51.5 | 65.5 | 53.3 | 55.0 |
| | | $1.3\times10^3$ | 66.4 | 71.1 | 39.2 | 58.6 | 47.6 | 52.1 |
| | | $2.7\times10^3$ | 73.9 | 75.1 | 60.4 | 68.9 | 57.7 | 57.9 |
| | | $5.4\times10^3$ | 73.0 | 74.6 | 57.6 | 67.9 | 56.2 | 57.1 |
| Resize | B/32 | $7.9\times10^1$ | 51.7 | 64.0 | 12.6 | 26.7 | 15.9 | 31.6 |
| | | $1.9\times10^2$ | 57.9 | 66.2 | 17.2 | 30.1 | 20.0 | 33.8 |
| | | $3.9\times10^2$ | 61.6 | 67.4 | 20.7 | 32.6 | 23.5 | 34.6 |
| | | $7.9\times10^2$ | 63.6 | 68.6 | 22.8 | 35.0 | 25.3 | 36.1 |
| | B/16 | $2.6\times10^2$ | 65.4 | 69.2 | 27.4 | 43.9 | 28.7 | 38.7 |
| | | $6.6\times10^2$ | 60.4 | 66.5 | 20.4 | 36.8 | 23.6 | 35.3 |
| | | $1.3\times10^3$ | 69.5 | 72.5 | 36.2 | 51.5 | 34.2 | 42.1 |
| | | $2.6\times10^3$ | 67.7 | 71.0 | 32.5 | 48.6 | 32.0 | 40.5 |
| | L/16 | $5.4\times10^2$ | 70.0 | 73.9 | 38.5 | 59.9 | 35.3 | 45.6 |
| | | $1.3\times10^3$ | 65.1 | 71.5 | 28.2 | 51.6 | 29.3 | 41.5 |
| | | $2.7\times10^3$ | 73.2 | 76.0 | 47.3 | 65.5 | 39.8 | 48.8 |
| | | $5.4\times10^3$ | 72.3 | 75.3 | 44.3 | 64.1 | 38.8 | 48.2 |

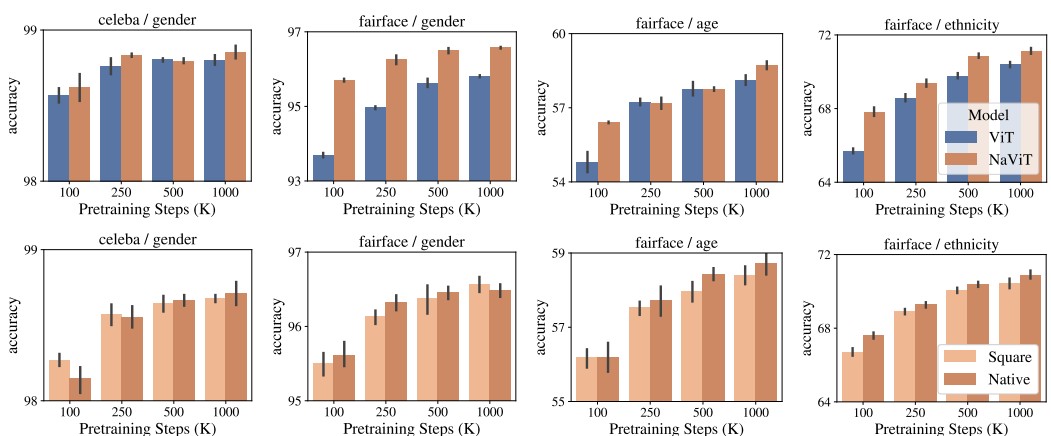

**Figure 22:** Summary of results of evaluating the accuracy of annotators trained on fairness-related signals using either NaViT-L/16 or ViT-L/16. TOP: NaViT offers better representations that improve the accuracy of annotators. BOTTOM: Using native aspect ratios in NaViT results in a higher performance when compared to resizing images to squares.

we emphasize that while NaViT improves the annotation accuracy in these tasks, care must be taken in such situations since classifiers can be inaccurate and lead to a broad categorization of people that misidentifies real identities. We encourage readers to delve into the comprehensive work outlining such potential risks, e.g. [58, 59], for further insight. In assessing the technical capabilities of NaViT, our intent is not to promote or encourage their application in inappropriate contexts. Rather, our objective is only to illuminate these technical findings for scenarios where they may be considered beneficial, such as when measuring the level of diversity in a dataset or auditing/mitigating biases in predictive models. We strongly advocate for responsible AI use, maintaining that the benefits of technological advancements should not overshadow the importance of user safety and privacy. AI tools, including

ours, should always be deployed judiciously, with a keen awareness of potential risks and a commitment to avoiding harm.

# I  Evaluation on model-vs-human OOD datasets on different resolutions

Just like NaViT, human visual perception works across flexible aspect ratios and resolutions (just imagine how strange the world would look like if we could only see it through a $224 \times 224$ pixel window!). We investigate how the ability to cope with variable resolutions affects performance on "model-vs-human", a benchmark of 17 challenging datasets [60].[1] For this purpose, we replicate the setup from Figure 6, but instead of evaluating ImageNet accuracy, we evaluate OOD accuracy on the model-vs-human benchmark.

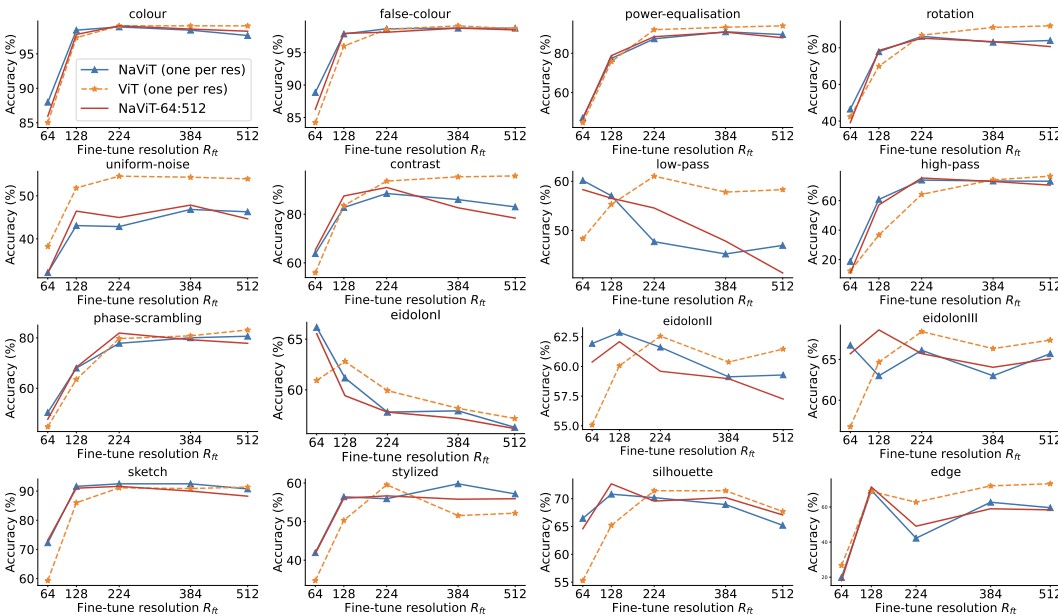

**Figure 23:** OOD accuracy on "model-vs-human" datasets across different fine-tuning resolutions. A single NaViT model trained on varying resolutions (red) performs roughly on par as fine-tuning one NaViT model per test resolution (blue). The ViT baseline (orange) is mostly worse than NaViT models for lower resolutions and mostly better for higher resolutions.

This corresponds to testing JFT B/16 models finetuned on ImageNet at various resolutions. The test dataset has a fixed $224 \times 224$ square resolution; thus we resize the test images to fit each model's fine-tuning resolution. Note that using square images has been standard practice designed for convolutional networks, but NaViT models no longer require square input, thus existing benchmarks are not tailored to those new possibilities. For datasets with multiple difficulty levels (such as different levels of blur), we average performance across levels while excluding levels that are too easy (not OOD) or too hard (human performance is close to chance), which follows the approach of **(author?)** [60] as explained in their "Appendix G Benchmark scores".

To ensure a fair comparison, we use models that are compute-matched for pretraining, and during fine-tuning, compute and data are identical for all models. The results of our comparison are shown in Figure 23. Overall, a single NaViT model trained on varying resolutions (red) performs roughly on par with fine-tuning one NaViT model per test resolution (blue).

The ViT baseline (orange) is mostly worse than NaViT models for lower resolutions and mostly better for higher resolutions. It may be worth noting that ViT models have a bit of an advantage in this comparison since they are fine-tuned on square images, whereas NaViT models are fine-tuned on flexible resolution images (preserving the image's aspect ratio) that have the same number of pixels, while not necessarily being square.

---

[1]For the purpose of our comparison, we exclude the "cue-conflict" dataset from the OOD evaluation, since there is no objective ground truth class in the case of images with a texture-shape cue conflict.

