# OpenReview forum: "Patch n’ Pack: NaViT, a Vision Transformer for any Aspect Ratio and Resolution"
_NeurIPS.cc/2023/Conference — NeurIPS 2023 poster_

### Official Review · Reviewer_SsJZ · 2023-06-28

**Soundness:** 3 good
**Presentation:** 3 good
**Contribution:** 3 good
**Rating:** 6
**Confidence:** 4

**Summary:**

This paper proposed an efficien training technique for Vision Transformers, called Patch n’ Pack. Specifically, it packs multiple images of various input resolutions into a single sequence as a batch exmaple. Furthermore, based on the modified architecture, the authors proposed NaViT. By combining Patch n’ Pack and NaViT, the authors conducted extensive experiments on JFT-4B datasets as well as a few downstream tasks. Overall, the method achieves great effciency for pretraining, and the model achieves better accuracy for differnet image resolutions at inference time.

**Strengths:**

1. Adapting general-purpose Transformers into different input image resolutions is a fundamental research problem. Therefore, the technique that proposed in this paper is important to the community. And it works pretty well on both pretraining and downstream tasks.
2. The experiments are comprehensive. The efficiency gain during pretraining is impressive.
3. This paper is easy to follow. The overall presentation is of great quality.

**Weaknesses:**

1. Packing examples into a single sequence during training is not new in the literature. However, it's technically new for ViT training.
2. Despite the performance, pretraining on JFT-4B is very expensive, making it difficult for the subsequent works to follow up and compare with. It would be better for the authors to include experiments of pretraining on ImageNet-1K.
3. It is not clear how the memory cost would be under the proposed Patch n’ Pack.


**Questions:**

It would be better for authors to add additional experiments for pretraining on ImageNet-1K and compare with FlexViT. Please also include memory consumption report during pretraining.

**Limitations:**

Yes

---

> ### Author Rebuttal · Authors · 2023-08-09
>
> Many thanks for time spent reading the paper and your critique and comments; we're glad you appreciated the significance of the research problem, clarity of writing, and the strength of results.
>
> We address here the mentioned weaknesses.
>
> 1. `Packing examples into a single sequence during training is not new in the literature. However, it's technically new for ViT training. `
>
> We agree, and have cited prior work accordingly. While this is a substantial technical change for Vision modeling, we however believe we contribute more still, by showing it unifies a number of approaches recently developed in computer vision literature (variable aspect ratio and token dropping), and enables new approaches which could not be explored before due to the restrictive need for constant sequence length per image.
>
> Alongside this, we have introduced variable resolution as a method for significantly speeding up training while producing more flexible models, and novel positional embedding schemas which enable generalization to larger resolutions. We think there's more here than just sequence packing, and believe as you pointed out there's a lot which would be of interest to the academic community.
>
> 2a. `Despite the performance, pretraining on JFT-4B is very expensive, making it difficult for the subsequent works to follow up and compare with.`
>
> This is a valid concern. We first note that other works demonstrating methodological improvements on JFT, such as the original Vision Transformer paper, opened the door to a plethora of research done at smaller scale (even experiments in the original Transformer paper were also at large scale in NLP). We believe the findings here are transferable to smaller datasets, open the door to additional innovation at smaller scales, and that this will therefore useful at many scales.
>
> That being said, based on Figure 1, the largest relative improvements seen are at smaller training schedules, which is promising for smaller scale research. We also note that we showed NaViT's techniques are useful during downstream finetuning, which are a much lower compute setup.
>
> `It would be better for the authors to include experiments of pretraining on ImageNet-1K.`
> That being said, we have been working on pre-training with public datasets, in order to develop a more reasonably reproducible setup and open-source models. Prior work [1] has shown that performant ViT models  pre-trained on ImageNet1k and ImageNet21k necessitate careful augmentation and regularisation. However, many of these techniques are very tailored to square images; for example, it’s unclear how mixup generalises to images of different sizes and shapes. It therefore needed more work than anticipated to adapt these techniques to NaViT.
>
> That being said, we ran some experiments, and when training NaViT-B/16 on ImageNet-1k, we can already match ViT’s performance with less than half the compute budget; see [this figure](https://i.imgur.com/KN3EOly.png) for initial results. This is work in progress, which we aim to finish for open-sourcing code around the conference, but cannot promise that it will be ready for the camera ready deadline. Nonetheless, as discussed, learnings from JFT have been known to transfer well to new setups, and trigger further research at smaller scales. Thus we believe the paper in its current form is still of significant value to the wider academic community.
>
> [1] How to train your ViT? Data, Augmentation, and Regularization in Vision Transformers, Steiner et Al, TMLR2022
>
> 3. `It is not clear how the memory cost would be under the proposed Patch n’ Pack.`
> Regarding memory costs, we have an initial discussion on this in Figure 4, though we appreciate in hindsight this is a bit abstract.
>
> Memory-wise, NaViT generally compares favorably. Assuming identical architecture and batch size, memory cost for both approaches is controlled by the sequence length. For ViT, this is set by the resolution. For NaViT, this is controlled by the maximum resolution we want to support. Assuming we keep NaViT's maximum resolution equal to the equivalent ViT baseline, which we would do anyway, they have identical memory costs.
>
> Concretely, if we train ViT-B/16 at resolution 384 on ImageNet-1k, it would have sequence length 576. An equivalent NaViT-B/16 could train on resolutions ranging from 64 to 384; it would have the same sequence length of 576  (and therefore the same memory cost), but it would fit in expectation almost two times as many images. At the same memory cost, NaViT will fit more images (or conversely, it can fit the same batch size, at a smaller memory cost).
>
> We believe we may have originally overstated the memory costs of NaViT, so we will clarify this in the paper.
>
> Many thanks again for time spent on this review; we hope this and the updated manuscript clarifies some aspects you raised.

---

### Official Review · Reviewer_z6Gn · 2023-07-03

**Soundness:** 3 good
**Presentation:** 3 good
**Contribution:** 2 fair
**Rating:** 6
**Confidence:** 4

**Summary:**

This paper focuses on adapting the computer vision model to flexible usage. The authors stand from the ViT architecture and exploit its flexible sequence-based modeling to enable arbitrary resolutions and aspect ratios. The proposed NaViT could benefit the downstream tasks of object detection, image, and video classification. Evaluations on typical ViT tasks show the performance on different downstream tasks.

**Strengths:**

Exploiting the flexible sequence-based modeling of ViT models is interesting. This paper uses a simple but effective idea to make the image preprocessing match arbitrary resolutions and aspect ratios. The idea is motivated by convincing preliminary experiments. The performance evaluation also presents useful insights into utilizing NaViT’s property.

**Weaknesses:**

The architecture design of NaViT and its essential components to extract visual features could be introduced to help the readers better understand the techniques.

Although the authors claim the proposed NaViT can be applied to different downstream tasks, the experiments are mainly based on high-level tasks of classification and detection. I am interested in NaViT’s generalization to more low-level tasks, such as super-resolution with arbitrary scales and pixel-level segmentation.

The additional overhead of introducing NaViT as the image preprocessing could be reported.

**Questions:**

Please see the suggestions in the weakness part.

**Limitations:**

None.

---

> ### Author Rebuttal · Authors · 2023-08-09
>
> Many thanks for your review and critiques; we have made a few updates to the manuscript based on your feedback, and hope it helps address some of the mentioned weaknesses.
>
> We will discuss them in more detail now:
>
> 1. `The architecture design of NaViT and its essential components to extract visual features could be introduced to help the readers better understand the techniques. `
>
> Aside from positional embeddings, the model architecture is functionally identical to the Vision Transformer. Given its widespread usage, we didn’t repeat ViT's architecture design and instead showed the differences introduced by our approach in Figure 2. We made this point more clear on the revised version of the paper and more explicitly directed readers to the Vision Transformer paper for further details on the architecture.
>
> 2. `Although the authors claim the proposed NaViT can be applied to different downstream tasks, the experiments are mainly based on high-level tasks of classification and detection. I am interested in NaViT’s generalization to more low-level tasks, such as super-resolution with arbitrary scales and pixel-level segmentation.`
>
> Good point! In the meantime, we have explored semantic segmentation on ADE20k, finetuning the L/16 models presented in Figure 1. The benefits of NaViT transfer naturally to this setting; for example, NaViT finetuned at resolution 384 outperforms ViT finetuned at resolution 512, while finetuning twice as fast (e.g. see [this figure](https://i.imgur.com/hQK7DCq.png) for segmentation results). We have added this figure and discussion of the results to Section 3.6 of the updated manuscript.
>
> 3. `The additional overhead of introducing NaViT as the image preprocessing could be reported`
>
> We assume the reviewer refers to the cost of image preprocessing. During training, as is common, the data preprocessing occurs on CPU. Typical for deep learning models at this scale, preprocessing is significantly faster than the time for a single training step (models are not input bound), and therefore there we observed no training time overhead of using Patch 'n' Pack.
>
> After training is complete, the resulting model is architecturally the same as ViT, except that it performs better, and generalizes better to new image sizes. But there is no additional overhead of running the model compared to an equivalent sized ViT.
>
>
> We hope these comments and the updated manuscript help address the mentioned concerns, and thank the reviewer again for their time!

---

> > ### Comment · Reviewer_z6Gn · 2023-08-20
> > **Post-rebuttal comments**
> >
> > Thanks for answering the questions. My concerns have been addressed.
> > I think it is a good paper and I have increased my rating.

---

### Official Review · Reviewer_CXkP · 2023-07-03

**Soundness:** 2 fair
**Presentation:** 3 good
**Contribution:** 2 fair
**Rating:** 5
**Confidence:** 4

**Summary:**

This passage discusses the common practice of resizing images to a fixed resolution before processing them with computer vision models, which is not optimal. The author introduces a new model called NaViT (Native Resolution ViT) that takes advantage of flexible sequence-based modeling and allows for processing inputs of arbitrary resolutions and aspect ratios with adaptive positional embeddings. NaViT uses sequence packing and token drop which improves training efficiency for large-scale supervised and contrastive image-text pretraining. The author believes that NaViT represents a promising direction for ViTs and offers a departure from the standard input and modeling pipeline used by most computer vision models, which rely on CNN-designed approaches.

**Strengths:**

- The authors present a simple method that significantly enhances the training efficiency of the vanilla ViT model, as evidenced by the results displayed in Figure 1. The observed improvements are noteworthy and suggest potential for practical application.
- The authors also make a compelling argument, supported by Figure 3, that the conventional practice of resizing or padding images to a fixed size, which has been historically associated with convolutional neural networks, is flawed. Specifically, the authors demonstrate that both resizing and padding can lead to suboptimal performance and inefficiency, respectively.
- While the overall algorithms are relatively straightforward, they are clearly depicted in Figure 2.

**Weaknesses:**

- The authors have presented a comprehensive set of experiments to demonstrate their results. However, I would like to raise a concern regarding the absence of comparison with the related works discussed in Section 4. This omission makes it challenging to evaluate the actual improvements over the baseline methods in a fair manner.

- To address this issue, I strongly encourage the authors to provide a detailed discussion on the primary contributions of their proposed methods. It appears that the example packing technique has already been thoroughly discussed in Efficient Sequence Packing [1], and one could simply replace the word tokens with image patches to form the proposed method. Moreover, apart from the example packing, the main difference between "Patch n'Pack" and Pix2struct [2] seems to rely solely on the construction of positional embedding. Additionally, it is worth noting that the most recent work [3] also aims at mix-resolution tokenization. While the proposed method may have some differences from existing works, the current manuscript fails to clearly establish the novelty of this paper.

### Reference:
- [1] Efficient sequence packing without cross-contamination: Accelerating large language models without impacting performance. Submitted to ICLR2022
- [2] Pix2struct: Screenshot parsing as pretraining for visual language understanding. Submitted to ICLR2023
- [3] Vision Transformers with Mixed-Resolution Tokenization. CVPR2023w

**Questions:**

- The authors have made an attempt to demonstrate the relative improvements over competitive counterparts in Figure 10. However, the differences between factorized position embeddings and NeRF appear to be marginal, with a range of less than $\pm 0.2\\%$.
- I would appreciate further clarification on the statement "This enables variable aspect ratios, with resolutions of up to $R=P\cdot \text{maxLen}$". Given that Pix2struct only introduces positional embeddings with the size of $[\text{maxLen},\text{maxLen}]$, and "indexed with (x,y) coordinates of each patch", I am curious how the resolution can be enlarged to $P\cdot \text{maxLen}$ without patch embedding in the size of $P\times P$.
- The authors have discussed several position encoding methods, but I remain unconvinced that any specific formulation significantly contributes to the performance improvements over the others.
- Could the authors provide additional details on the settings of "the total number of pixels can be resampled while preserving the aspect ratio"? While I agree that "for NaViT a resolution of “128” has the same area of a square 128 x 128 image, but could be 64 x 256, or 170 x 96", I am curious as to why the "effective resolution" of $64 \times 256$ is larger than $128 \times 128$.
- The authors have thoroughly discussed the benefits of the variable resolution, sampling strategies, and token dropping. However, none of these concepts are first proposed in this work. While the details may differ from existing works, such as the resolution sampling strategy, the fundamental idea remains the same. Therefore, the novelty of this paper appears to be limited.
- The relationship between "Pack n'Patch" and "Self-attention cost" appears to be weak, as the packing of multiple patches from different images into a single sequence does not increase the cost compared to the original single sequence scheme. We kindly request the authors to provide further clarification on why the attention overhead should vary with the number of patches packed.

**Limitations:**

I found no potential negative societal impact of their work.

---

> ### Author Rebuttal · Authors · 2023-08-09
>
> Thank you for the detailed review and comments.
>
> We first address the weaknesses section, and in particular concerns around lack of novelty.
>
> *Weaknesses*
> Sequence packing, variable aspect ratio, and token dropping are indeed not new concepts, and have independently been studied. This is not true for variable resolution; multiresolution models are common in the literature, typically for dense prediction tasks, but to our knowledge this is the first work to explore variable resolution to speed up training (we also note that your reference [3] was uploaded one month after the NeurIPS submission deadline).
>
> As well as this, our major contributions are:
> * Introducing sequence packing to computer vision (Patch n’Pack), and demonstrating that it combines very well with multiple recent advancements in computer vision (token dropping and variable aspect ratio).
> * Introducing variable resolution as a way to speed up training.
> * Demonstrating that Patch n’ Pack enables a far richer design space for these computer vision techniques than was previously possible (e.g. sampling per-image resolutions and token dropping rates), and showing initial benefits from doing so.
> * Introducing positional embedding methods which demonstrably improve generalization to new resolutions.
>
> We believe this will be of great interest to the academic community, change the de-facto non-packed method of training vision models, and meets standards of novelty for publication.
>
> *Questions*
>
> `...the differences between factorized position embeddings and NeRF appear to be marginal, with a range of less than 0.2%. `
>
> `...remain unconvinced that any specific formulation significantly contributes to the performance improvements over the others.`
>
> Figure 10-left presents the difference between positional embedding methods, and the gaps between different approaches are indeed small.
> However, this is for the evaluation in the “in-distribution- setup”, where the train and test resolutions are in a similar range.
> The differences between different methods are more significant in out-of-distribution resolution evaluations (Figure 10 right); this is the main reason we propose the alternative approaches.  We added a paragraph to the positional embedding section to clarify this.
> The benefits of our approach will hopefully also become clearer below in our response to the next question.
>
> `...further clarification on the statement "This enables variable aspect ratios, with resolutions of up to R=P. Given that Pix2struct...`
>
> During training there is a maximum sequence length. As an example, say it was 9 tokens; with an example patch size of P = 10, this corresponds to a 30 x 30 image (or 10 x 90, or 20 x 40 with one padding token, etc).
>
> For Pix2struct we would correspondingly `max_len = 9`. It uses a fixed grid of 2D embeddings of size `9 x 9`. During training, only images with a sequence length ≤ 9 are seen. This means only positional embeddings where $x \times y \leq 9$ are actually trained; the rest remain randomly initialized. At inference time, given a larger image, most combinations of $(x, y)$ will not be trained as any larger image will have some $x, y$ where $x\times y > 9$. The pix2struct approach therefore cannot generalize to larger images; we have prepared [this diagram](https://i.imgur.com/8MWdkZK.png) to hopefully also help.
>
> Our factorized approach uses separate positional embeddings for X and Y coordinates. In the above example, each of these positional embeddings is of length `9`. All positional embeddings are used during training, which means when running inference on larger images, no untrained positional embeddings are used. It can e.g. generalize to a 40 x 40 image, as it saw 20 x 40, and 40 x 20 during training, even though a 40 x 40 image would never be seen during training as its sequence length of 16 exceeds the max length of 9. [This diagram](https://i.imgur.com/hjKnvSB.png) visualises this.
>
> We hope this clarifies the setup!
>
> `...provide additional details on the settings of "the total number of pixels can be resampled while preserving the aspect ratio...`
>
> By effective resolution, we mean: $\sqrt(\mathtt{num\textunderscore pixels})$.
> The effective resolution of 64 x 256 is the same as 128 x 128, as they both have the same number of pixels (64 * 256 = 128 * 128 = 16384).
>
> What we discussed there is the difference in approaches to resizing. Typical computer vision pipelines would squash an image to square, and then resize to desired resolution $R$.
> We instead resize the image such that it has (roughly) $R^2$ pixels. This has the same number of pixels, but retains the aspect ratio. It is important to have the same number of pixels, as this controls the number of patches and therefore the compute cost, and allows for comparable evaluations.
>
> `...Therefore, the novelty of this paper appears to be limited.` We hope this question is well addressed by the first response!
>
> `The relationship between "Pack n'Patch" and "Self-attention cost" appears to be weak, ...`
> Self attention cost scales quadratically with sequence length $n$.
> With $B$ images, each of sequence length $n$, the cost is roughly $O(Bn^2)$.
> If instead we pack $k$ images per sequence, increasing sequence length to $kn$ but processing a smaller batch of size $\frac{B}{k}$, the cost would be $O(\frac{B}{k} \times (kn)^2) = O(Bkn^2)$.
> This extra factor in the cost is broadly speaking the concern around the self attention cost of Patch n’ Pack.  However, we note two things:
> * The self attention cost of a transformer model becomes an increasingly small proportion of overall costs as transformers scale up. This is what is shown in Figure 4.
> * We don’t actually have to increase the sequence length to benefit from NaViT. With the same max sequence length, NaViT can mix in images at smaller resolutions (less tokens than n) and still enjoy the increased throughput.
>
> We hope we have addressed some of the concerns and the updated manuscript is to a satisfactory satndard!

---

> > ### Comment · Reviewer_CXkP · 2023-08-20
> > **Re: Rebuttal by Authors**
> >
> > Thank you for the detailed feedback. After carefully considering the comments from the other reviewers, I still have some reservations about the novelty of the proposed technique. While "explore variable resolution to speed up training" may be new in the literature on vision transformers, I am not fully convinced that combining sequence packing, variable aspect ratio, and token dropping should be considered a significant contribution to the field.
> >
> > However, I must acknowledge that the authors have addressed all of the concerns raised in the rebuttal, and there are no apparent flaws in the proposed method. Therefore, I raise my previous rating to borderline.
> >
> > I would like to note that the authors claim that "Introducing variable resolution as a way to speed up training" is one of the major contributions of this paper, and that this technique has been discussed on CNN-backbone as a group of Resolution-level Dynamic Networks.
> >
> > Finally, I would encourage the authors to provide further reflection on the response to the last question in the final version of the paper. Without this clarification, it may be difficult for readers to fully understand the trade-offs involved in using the proposed method, and the paper may not be as impactful as it could be.

---

### Official Review · Reviewer_WGZP · 2023-07-04

**Soundness:** 4 excellent
**Presentation:** 3 good
**Contribution:** 3 good
**Rating:** 6
**Confidence:** 3

**Summary:**

The authors propose to use example packing to train ViTs, where training examples of various lengths are packed into a single sequence. This requires a few straightforward architecture changes, including modified attention masking, pooling, and positional embeddings. This scheme allows for some interesting ideas, such as variable image resolutions during training, variable token drop rates, and adaptive inference time computation.

Extensive experiments in the paper show that NaViT results in more efficient training and finetuning (in terms of TPU hours) and a better compute-performance tradeoff at inference time. Further analysis shows that mixed resolution training is beneficial to model performance, that the time-varying token drop ratio allowed by the model can improve results when the correct schedule is used, and that the proposed factorized positional embeddings can offer very good generalization to aspect ratios and resolutions unseen during training. Finally, additional experiments show promising behavior with respect to calibration, fairness, object detection, and video classification.

**Strengths:**

Overall, this is a very comprehensive paper. The number of experiments and the aspects of the proposed models performance evaluated is impressive. The authors touch on a large number of relevant properties of the model, including calibration stability, positional embedding ablations, and out of distribution performance. Additionally, results are quite positive, showing promising results on image classification, object detection, video classification, and training efficiency. Finally, the appendix is thorough, and gives enough details to accurately reproduce all experiments from what I can see.

**Weaknesses:**

According to appendix section B.1, classification experiments seem to be "compute-matched." Are there any experiments analyzing the asymptotic performance between ViT and NaViT, or can the authors discuss this? The experiments in the paper seem inconclusive on this point.

The question of compute matching also applies to downstream experiments. For example, in the fairness analysis, are compute matched ViT and NaViT being compared? If so then the conclusion from Appendix H that "native image resolution improves the performance of fairness signal annotation" may not hold.

NaViT seems to benefit from training at the original aspect ratio, and at variable resolutions. However, could either of these benefits be achieved through scale and crop data augmentation? What data augmentation is used for ViT? Additionally, I was under the impression that  image stretching or shearing could be useful as a data augmentation. Could the authors please discuss this?

Where are the contrastive results? I might have overlooked something, but I can't find zero-shot imagenet or COCO image-text retrieval results as discussed on line 154.

Minor typos:
- Figure 2 typo: "image 2" is repeated twice in the "data processing" part
- Figure 9 is before Figure 7 and 8
- It seems that a lot of (if not all) appendix section references are incorrect. Examples of these typos occur in line 152, line 158, line 339, and line 120.

**Questions:**

See weaknesses section, thank you.

**Limitations:**

I think the "limitations" section from the appendix could be more forthcoming and candid. It mostly just says "NaViT" is a great idea, but we didn't get to apply it to all the tasks we wanted to. It could benefit from a more honest discussion of the method's limitations, such as compute overhead from packing or additional architectural changes needed to support example packing.

---

> ### Author Rebuttal · Authors · 2023-08-09
>
> Firstly, many thanks for your time and detailed feedback. It was very useful and constructive, and we made multiple improvements to the manuscript based on your comments. Please see below for some detailed responses to the weaknesses you pointed out.
>
> 1. `Compute matching and asymptotic performance`
>
> This is a good point; based on current experiments, we believe a lot of the observed performance benefits are due to speeding up training with NaViT (i.e. seeing more images in less time).
> We introduce [this figure](https://i.imgur.com/Rpn4GBU.png) which augments Figure 1. In the center, we can show that per-token seen, NaViT trains much faster than ViT (and per TPU-hour, left), and this is because Patch 'n' Pack enables NaViT to see tokens from more unique images (figure, right).
>
> Comparison of asymptotic performance can be difficult. First, when training on these large pre-training on large datasets, it is prohibitively expensive to train to convergence, so more training always yields improvements. Therefore it is common for foundation models to compare at a fixed (often large) training budget as we show above. Even at the largest budget ( a large 2000+ TPU chip hours), NaViT is still far ahead.
>
> Further, in the infinite compute limit, for a given number of images seen, it is best to train on purely large resolution, without token dropping. This is not a setting that we explore since it substantially slows down training. Asymptotically, we therefore expect the token dropping and mixed resolutions to eventually hurt NaViT, although we believe that it would be very expensive to approach this regime.
> We have added a new section (Appendix B.5) to the appendix with this figure, discussing asymptotic performance, and added pointers to it in the introduction, in the updated version of the paper.
>
> 2. `Compute matching downstream experiments`
> It would indeed be unfair to compare non-compute matched models, since pre-training on large datasets (like JFT) for longer always improves performance. Downstream experiments were all done using models pre-trained for the same amount of time, and evaluated at the same effective resolution (number of tokens) - i.e. compute matched for both training and eval cost. Specifically, *all downstream experiments used the top-rightmost points in Figure 1 (ViT-L/16 and NaViT-L/16)*. We have added a paragraph to the experimental setup to make this point clear.  Many thanks for raising this.
>
> 3. `NaViT seems to benefit from training at the original aspect ratio, and at variable resolutions. However, could either of these benefits be achieved through scale and crop data augmentation?`
>
> We do not believe aspect-ratio conservation provides an augmentation effect. It may interact with other data augmentations; e.g. inception-cropping samples a rectangular bounding box, and then squashes it back to a square. This may work better with aspect-ratio preserving models.
>
> Variable resolution and token dropping could indeed act as data augmentation; we did not conclusively demonstrate this in the paper, since in the large-scale pre-training regime we explored typically doesn’t benefit significantly from data augmentation.
>
> We are actively exploring this in smaller-scale settings; this is however quite an undertaking. Many of the data augmentations/training techniques which enable training ViT on small-scale models, such as mixup, require some rethinking when dealing with non-square or variable resolution images. We therefore leave research on data augmentation for Native resolution vision models to future work.
>
> 4. `What data augmentation is used for ViT? Additionally, I was under the impression that image stretching or shearing could be useful as a data augmentation. Could the authors please discuss this?`
>
> We follow the original Vision Transformer paper, and follow-ups; models are trained on images with resolution 224×224, with inception crop followed by random horizontal flipping. Pre-training on JFT and large image-text datasets does not benefit much from heavy data augmentation as discussed above, and it isn’t common practice to use image stretching or shearing; this is also the case for other computer vision works with large datasets e.g. CLIP. We added a small section to the appendix  (Appendix B.4) with these details.
>
> 5. `Where are the contrastive results? I might have overlooked something, but I can't find zero-shot imagenet or COCO image-text retrieval results as discussed on line 154.`
>
> Apologies, we realise in hindsight these weren't clearly highlighted. A few of the experiments were performed in a contrastive setup, e.g. Figure 7, and the contrastive-pretrained models used to initialize the detection OWL-ViTs in Section 3.6, where zero-shot ImageNet accuracy was reported for both. We will make this more clear for the camera ready version.
>
> 6. `Minor typos` many thanks for noticing these, we have corrected them in the uploaded pdf!
>
> 7. `Honest limitations section`
>
> Thanks for raising this. We have updated the limitations section to:
> * Discuss compute overhead, referencing figure 4 and related discussions.
> * Touch on the complexities of implementations/architectural changes.
> * Mention what we discussed above about the unexplored data-augmentation angle.
> * Mention some limitations relating to small-scale experiments.
>
> Thanks again for your insightful review; we hope these have answered your questions, and that the updated manuscript reflects the addressed feedback.

---

> > ### Comment · Reviewer_WGZP · 2023-08-19
> >
> > Thank you for answering my questions and updating the manuscript where appropriate. The response has cleared up some of my confusion and has resolved all of my concerns satisfactorily.
> >
> > In addition, I have read the other reviews and corresponding rebuttals, and I am of the opinion that the author has given reasonable responses to points brought up by the other reviewers. In particular, two reviewers are concerned about the novelty of the work. On the contrary, I agree with the authors that while sequence packing, variable aspect ratio, and token dropping have been proposed before, their combined use in this work is novel (i.e. the fact that sequence packing can quite naturally enable variable resolution or aspect ratio training and dynamic token dropping).
> >
> > Overall, I still think the paper is a good submission and I want to keep my rating as is. I would be happy to discuss additional questions or points that the authors or other reviewers may have.

---

### Author Rebuttal · Authors · 2023-08-09

We thank the reviewers for their time and valuable comments. We were happy to hear that they found the paper “*very comprehensive*”, leading to “*impressive performance*” (WGZP), making a “*compelling argument*” that we need to go beyond fixed-size resolutions (CXkP), based on an “*interesting idea*” and backed up with “*useful insights*'' (z6Gn) that leads to “*impressive efficiency gains*” as a result of solving a “*fundamental research problem*” and, finally, that the paper is “*easy to follow*” due to its “*great presentation quality*” (SsJZ).

We have made multiple modifications based on the feedback, including clarification of text, fixing typos, adding semantic segmentation results, and adding/deepening discussions on asymptotic performance and limitations.

---

### Decision · Program_Chairs · 2023-09-21

**Decision:**

Accept (poster)

**Comment:**

The paper proposes a method for training ViTs with heterogeneously-sized images. Reviewers were generally favorable on the paper, but had some questions about the experimental setup, relation to prior work, compute costs, and the applicability of the method to other tasks. The authors submitted detailed responses that addressed the majority of these concerns, and in the end all reviewers were in favor of accepting the paper.

The one outstanding concern is the relationship between the proposed method and prior work raised by Reviewer CXkP. While there is certainly similarity between the proposed method and prior methods, the AC is sympathetic to the arguments of the authors regarding novelty, and believes that this paper will have value to the community.

The authors are encouraged to update the camera-ready version of the paper in response to the suggestions of the reviewers.